# Identification of RING E3 pseudoligases in the TRIM protein family

Jane Dudley-Fraser [1], Diego Esposito [1], Katherine A. McPhie [1], Coltrane Morley-Williams[1], Tania Auchynnikava[2] & Katrin Rittinger [1] ✉

TRIpartite Motif (TRIM) family proteins have diverse roles across a broad variety of cellular functions, which are largely presumed to depend on their ubiquitin E3 ligase activity, conferred by a RING domain. However, recent reports have shown that some TRIMs lack detectable ubiquitination activity in isolation, despite containing a RING domain. Here, we present parallel in cellulo, in vitro, and in silico structure-function analyses of the ubiquitin E3 ligase activity and RING domain structural characteristics of the whole TRIM protein family. In-depth follow-up studies of this comprehensive dataset reveals a number of 'pseudoligases', whose RING domains have structurally diverged at either the homodimerisation or E2-ubiquitin interfaces, thereby disrupting their ability to catalyse ubiquitin transfer. Together, these data raise intriguing open questions regarding the unknown TRIM functions in physiology and disease.

The tripartite motif (TRIM) family of proteins comprises 77 known members in humans, which are implicated in diverse cellular activities across different tissues, including infection, differentiation, DNA damage and oncogenesis[1–3]. However, despite their links to a wide variety of pathologies, the mechanistic and regulatory features of the majority of TRIM family members remain poorly characterised[3,4].

At their N-termini, TRIMs contain a highly conserved TRIpartite Motif domain structure, made up of a RING domain, one or two B-box domains and a long coiled-coil domain (RBCC)[5]. The zinc co-ordinating RING and B-box domains are generally believed to confer ubiquitin E3 ligase activity and higher order oligomerisation, respectively, while the coiled-coil domain mediates anti-parallel TRIM homodimerisation[6]. The C-terminal domains (CTDs) of TRIMs, however, vary and facilitate diverse protein-protein interactions[7]. TRIM grouping into 12 sub-classes (I–XI and unclassified, Fig. 1a) is dictated by their CTD composition.

Although several different functions have been attributed to TRIMs, including transcriptional regulation and SUMOylation, the majority of studies have focussed on the RING-dependent ubiquitination of protein targets[8–11]. Some diverse examples include: the ubiquitination and degradation of cytoskeletal protein NF-L by TRIM2 to regulate axonogenesis during brain development[12]; TRIM21-mediated ubiquitination of intracellular antibodies to remove antibody-bound viruses[13,14]; and the ubiquitination of ATG7 by TRIM32 to drive autophagy and cellular adaptation to oxidative stress[15]. However, a number of recent studies have reported that certain TRIMs seem incapable of catalysing ubiquitination[16–18]. In some cases, in-depth structural analyses have highlighted key features of ubiquitin ligase-defective TRIMs, such as an inability of the RING to dimerise[6]. RING dimerisation is a key aspect of the catalytic mechanism of TRIM RING ligases, where, following the E1-mediated loading of ubiquitin onto an E2 enzyme (forming an E2-Ub thioester), the E2 binds one RING domain whilst the conjugated ubiquitin makes contacts with both RINGs in the dimer (Fig. 1b). This is required to stabilise the E2-Ub into a 'closed' catalytic conformation that is primed for subsequent ubiquitin discharge onto substrate lysine residues[19–23]. Dimerisation is mediated by the α helices N- and C-terminal to the 'core' of each RING domain that combine to form a four-helix bundle, with TRIMs lacking these helices unable to catalyse ubiquitination (e.g. TRIMs 3, 24, 28 and 33)[18]. It remains to be examined, however, how many RING-containing TRIMs might have structural features that are not compatible with canonical ubiquitination activity and, if not, whether they become active after specific cellular stimuli or simply retain an inactive RING domain and carry out alternative functions. Indeed, there are several RING-less TRIMs

[1]Molecular Structure of Cell Signalling Laboratory, The Francis Crick Institute, London, UK. [2]Proteomics Science Technology Platform, The Francis Crick Institute, London, UK. ✉e-mail: katrin.rittinger@crick.ac.uk

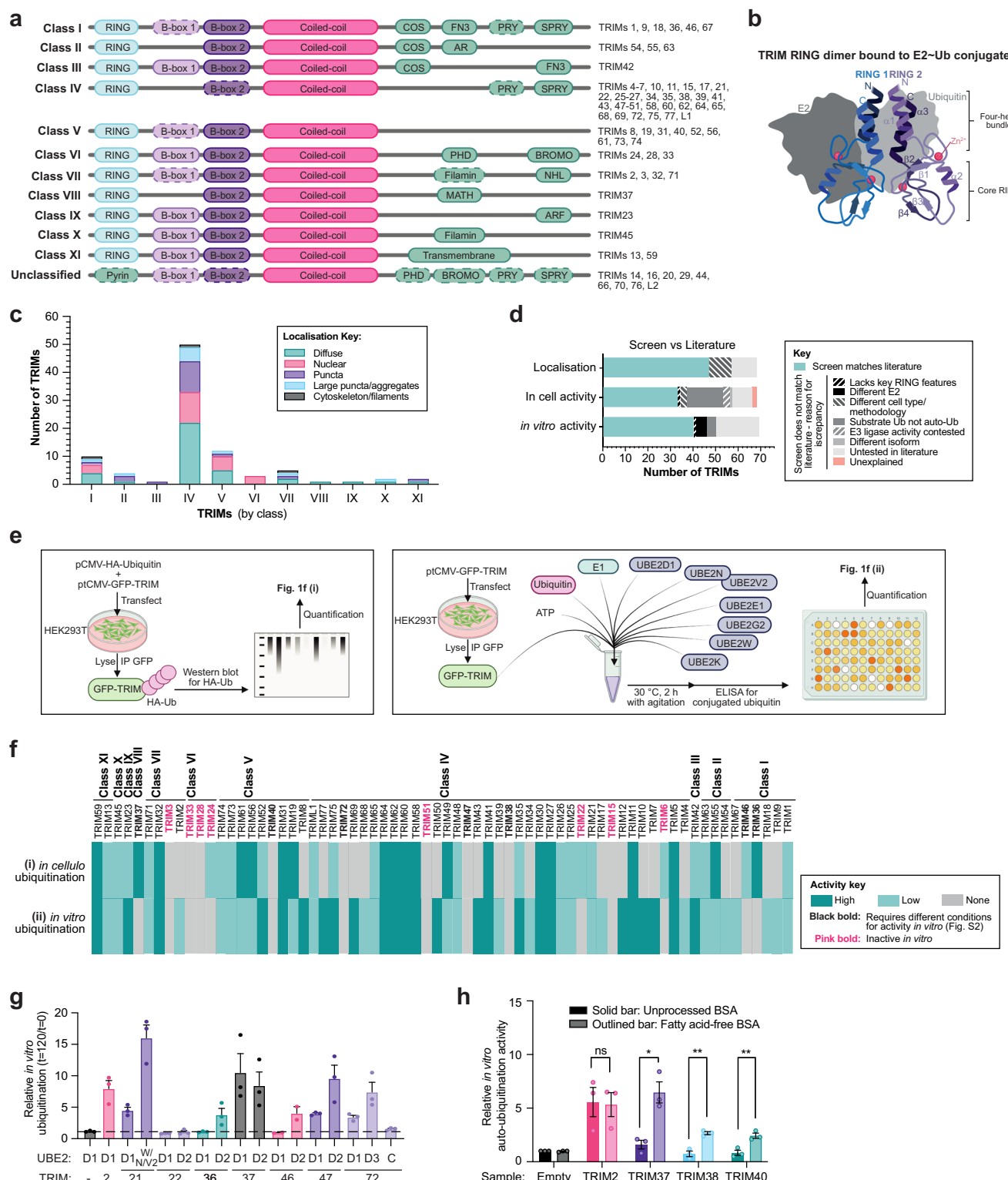

(Fig. 1a: 'Unclassified'), which can exert distinct regulatory functions via the BCC and C-terminal domains[24–26].

To understand how many TRIM proteins do not exhibit ubiquitin E3 ligase activity, here we carried out a family-wide analyses of ubiquitination in cells and in vitro. To put these results into a molecular context, we examined Alphafold2 and AlphaFold3 structural models of TRIM RING domains for features and residues of known catalytic importance, which revealed key structural discrepancies in TRIMs lacking ubiquitination activity in our analyses. Unexpectedly, we find

that the RING domains of a subset of class IV TRIMs have structurally diverged, thereby ablating their ubiquitin E3 ligase activity. Sequence variations in this group, which has previously been described to exhibit rapid evolutionary dynamics[2], have resulted in RING domains that either do not dimerise (as previously described for TRIMs 3, 24, 28 and 33) or do not functionally engage the E2-Ub conjugate. Furthermore, our study adds to existing literature that ligase-deficient TRIMs can interact with and negatively regulate ligase-proficient TRIMs, in the case of homologues TRIM51 and TRIM49. Functionally, these TRIMs

**Fig. 1 | Screening TRIM family for localisation and in-cell and in vitro auto-ubiquitination activities. a** Diagram depicting TRIM family protein domain organisation according to class, dashed lines indicate that the domain is omitted by some TRIMs in that class. **b** Diagram representing a canonical TRIM RING dimer bound to the E2-Ub thioester, with key structural features and $Zn^{2+}$-binding sites labelled. Created in BioRender. Dudley, J. (2025) https://BioRender.com/o30m956. **c** Quantification of localisation of GFP-tagged TRIM proteins transiently over-expressed in U2OS cells (Supplementary Fig. 1, n ≥ 3). **d** Graph representing the comparison of TRIM localisation and activities screens in this study with currently available TRIM literature, highlighting likely causes for any discrepancies (Supplementary Tables 1 and 2). **e** Diagrams depicting experimental design for in-cell ubiquitination assays (left) and in vitro auto-ubiquitination assays (right). Created in BioRender. Dudley, J. (2025) https://BioRender.com/q12b017. **f** Heat map representing mean ubiquitination (i) *in cellulo* (quantification of western blotting of HA-Ubiquitin relative to GFP-TRIMs immunoprecipitated from HEK293T cells. n = 4 independent experiments) and (ii) in vitro (reaction at 30 °C 120 min: 1 μM mix of E2 enzymes (see **e**), 1 μM UBA1, 50 μM ubiquitin, 3 mM ATP and immunoprecipitated GFP-TRIMs. HRP signal from conjugated ubiquitin-specific ELISA standardised relative to GFP signal detected by plate reader. n = 3 independent experiments). Dark and light teal: 'high' and 'low' relative auto-ubiquitination above threshold levels, respectively. Grey: lack of detectable ubiquitination above levels of GFP-empty negative controls. **g** Bar graph representing the quantification of ubiquitination western blots of in vitro ubiquitination assay reaction mixtures (t = 120/t = 0 min reaction time) of GFP-tagged TRIM proteins purified from HEK293T cells as described in (**f**(ii)), but using different E2 enzymes (UBE2C, UBE2D1, UBE2D2, UBE2D3, or a mix of UBE2N/UBE2V2/UBE2W) (n = 3). Circles: individual values, error bars: mean ± SEM (n = 3). **h** Bar graph representing the quantification of ubiquitination ELISA assay as described in (**f**(ii)), comparing sample preparation in buffer with unprocessed BSA (dark bars), and fatty acid-free BSA (light bars). Circles: individual values, error bars: mean ± SEM (n = 3, two-tailed *t*-tests, *P* values left to right: 0.911, 0.025, 0.007 and 0.007 (ns > 0.05, *<0.05, **<0.01, ***<0.001)). Source data are provided as a Source Data file.

exhibit contrasting effects on autophagic flux, which TRIM49 promotes but TRIM51 represses.

Herein we identify and characterise several examples of TRIMs that lack canonical ubiquitin ligase activity, which we dub 'pseudoligases'. These results highlight exciting opportunities to explore the hitherto unrecognised ubiquitination-independent cellular functions of TRIM family proteins. Moreover, these findings take us a step closer to tapping into their potential as drug targets or targeted protein degraders, with impacts across the many physiologies and pathologies where TRIMs have been ascribed roles.

## Results

### TRIM localisation in mammalian cells varies within and between classes

To analyse the properties of the TRIM family, a library of 68 RING-containing TRIMs tagged with eGFP was generated for expression in mammalian cells. TRIMs 53 and 57 were excluded as they are now considered to be pseudogenes[27], whereas murine isoforms Trim12 and Trim30 were included due to their close homology to TRIM5, a well-characterised player in viral restriction[28].

The TRIM library was expressed in U2OS cells to assess localisation by widefield microscopy (Fig. 1c, Supplementary Fig. 1). TRIM localisations observed in this study agree well with previous reports (Supplementary Table 1), with 11 exceptions, which are likely attributable to methodological differences (e.g. cell types, staining methods, or epitope tags) (Fig. 1d). Over half of TRIMs (54%) exhibit diffuse cytoplasmic localisation, whilst 39% form either small or large puncta, and 5% localise to filamentous cytoskeleton-like structures. Moreover, 32% of TRIMs localise to some degree to the nucleus, either exclusively (e.g. TRIM24) or in addition to another localisation pattern (e.g. TRIM11 is diffuse throughout both the cytoplasm and nucleus). Whilst this may allude to nuclear regulatory functions for these TRIMs, it may also reflect the inherent nuclear translocation tendency of the GFP tag itself[29].

Notably, however, we find that TRIM localisation correlates poorly with C-terminal domain classifications, with the exception of the nuclear localisation of all PHD-BROMO-containing TRIMs (class VI). Considering that TRIM C-terminal domains are generally considered to dictate substrate binding, it is intriguing that members of a given class do not localise to common types of subcellular structures.

### Auto-ubiquitination analyses in cells and in vitro indicate some TRIM proteins lack ubiquitin ligase activity

Next, we sought to assess TRIM ubiquitin E3 ligase activity on a family-wide scale. Given that ubiquitination substrates have not been validated for the majority of TRIMs, auto-ubiquitination was utilised as a proxy read-out for activity. Two methods were employed: a) in cellulo analysis via the immunoprecipitation (IP) of GFP-tagged TRIMs co-overexpressed with HA-tagged ubiquitin in HEK293T cells (treated with proteasome (MG132) and pan-deubiquitinase (PR619) inhibitors), with HA-auto-ubiquitination status then analysed by western blotting (Fig. 1e, f, Supplementary Fig. 2a); and b) in vitro ubiquitination assays conducted with GFP-tagged TRIMs IPed from HEK293T cells incubated with a reaction mix containing ATP and recombinant ubiquitin, E1 and a cocktail of seven E2s, designed to include the most common E2s used by TRIMs (UBE2D1, E1, G2, K, N/V2, W; see Supplementary Fig. 2b for condition optimisation), which was then analysed by a direct ELISA assay with the FK2 antibody specific for conjugated ubiquitin (Fig. 1e, f, Supplementary Fig 2b, c). These commonly used assays were selected as they are applicable at the scale required in this study and have complementary benefits and limitations. Whilst in cellulo analyses ensure the presence of any required accessory factors, they can be confounded by TRIM ubiquitination by other E3 ligases present in the cellular milieu and the possible preferential ubiquitination of available substrates by TRIM proteins over auto-ubiquitination. On the other hand, in vitro assays lack potential regulatory factors and subcellular localisation (e.g. membrane binding[30]), but benefit from a clear definition of experimental components. Additionally, neither assay can guarantee a full complement of all possible E2 enzymes. However, as no single approach is available to control for all these factors at present, the combination of these two assays allowed a first assessment of ubiquitination activity on a family-wide level and identified candidate TRIMs for more in-depth studies.

In cells, 27 of 68 RING-containing TRIMs did not show in cellulo ubiquitination above background levels (TRIMs 2–4, 7–10, 12, 15, 17, 18, 28, 33, 34, 38-40, 43, 47, 50, 51, 54, 67–69, 72 and 77), possibly indicating that these preferentially ubiquitinate substrates over auto-ubiquitination. Meanwhile, in vitro auto-ubiquitination was not detected for 15 TRIMs (TRIMs 3, 6, 15, 22, 24, 28, 33, 36–38, 40, 46, 47, 51 and 72) (Fig. 1f). These results are largely in line with activities previously reported in the literature, with the majority of discrepancies likely due to key differences in experimental conditions (Fig. 1d, Supplementary Table 2). For example, the in vitro ubiquitination assays carried out here reflect the findings of previous studies showing that TRIMs 3, 24, 28 and 33 lack auto-ubiquitination activity in isolation in vitro. Their deficient ligase activity was confirmed here by western blotting against in vitro ubiquitination assays, alongside TRIM2 as a positive control (Supplementary Fig. 2d). It is possible that these TRIMs may require additional factors, alternative E2s or post-translational modifications for ligase activity as has been previously described for some TRIMs (e.g. TRIM28 interaction with MAGEA3/6[31]), that they do not auto-ubiquitinate, or that they have an alternative function in cells.

Of the other RING-containing TRIMs that did not show ubiquitin ligase activity in in vitro assays, TRIMs 22, 36, 37, 47 and 72 have been described to have activity in previous studies with E2 enzymes that

were not included in our E2 cocktail (Supplementary Table 2). These TRIMs were therefore tested in in vitro reactions with alternative E2s and assessed by western blotting, with positive control reactions of TRIM2 with UBE2D1 and TRIM21 with the mixture of UBE2W/UBE2N/UBE2V2, as previously described[14,17] (Fig. 1g, Supplementary Fig. 2e). This experiment showed that TRIM36 functioned with UBE2D2, not D1, as did closely related family member TRIM46. Although TRIM47 and TRIM72 exhibited weak activity with UBE2D1, both had enhanced auto-ubiquitination with UBE2D2 and UBE2D3, respectively. TRIM22 auto-ubiquitination was not detected by either E2, which is investigated further later in this study. On the other hand, auto-ubiquitination activity of TRIMs 37, 38 and 40 was detected by western blotting but not by ELISA (Fig. 1h, Supplementary Fig. 2f). After detailed examination, it was determined that these TRIMs were disrupted by the lipid composition of BSA in the ELISA sample buffer, which was resolved by utilising a fatty acid-free BSA (Supplementary Fig. 2g). This may reflect the possibility of these TRIMs functionally interacting with lipids as,

interestingly, TRIMs 37 and 38 have been connected to lipid metabolism[32,33]. Therefore, although the ELISA offered high-throughput analysis, validation by western blotting proved crucial and was performed to confirm the findings throughout this study.

## In silico analysis of TRIM RING domains highlights structural differences in ubiquitination-deficient TRIMs

It is known that certain structural features are required for TRIM ubiquitin E3 ligase activity, such as RING dimerisation mediated by a four-helix bundle formed by α-helices located N- and C-terminal to the core RING domain (Fig. 2a)[17,18]. Therefore, to better understand the results of our family-wide analyses, models for the RING domains of all TRIM family proteins were generated using AlphaFold2 and aligned according to class (Fig. 2b, Supplementary Fig. 3)[34,35]. Detailed examination revealed key features for three of the potentially inactive class IV TRIMs: TRIM6 is predicted to lack the α2 helix in the core RING domain, a region known to both contact the E2 enzyme and stabilise a

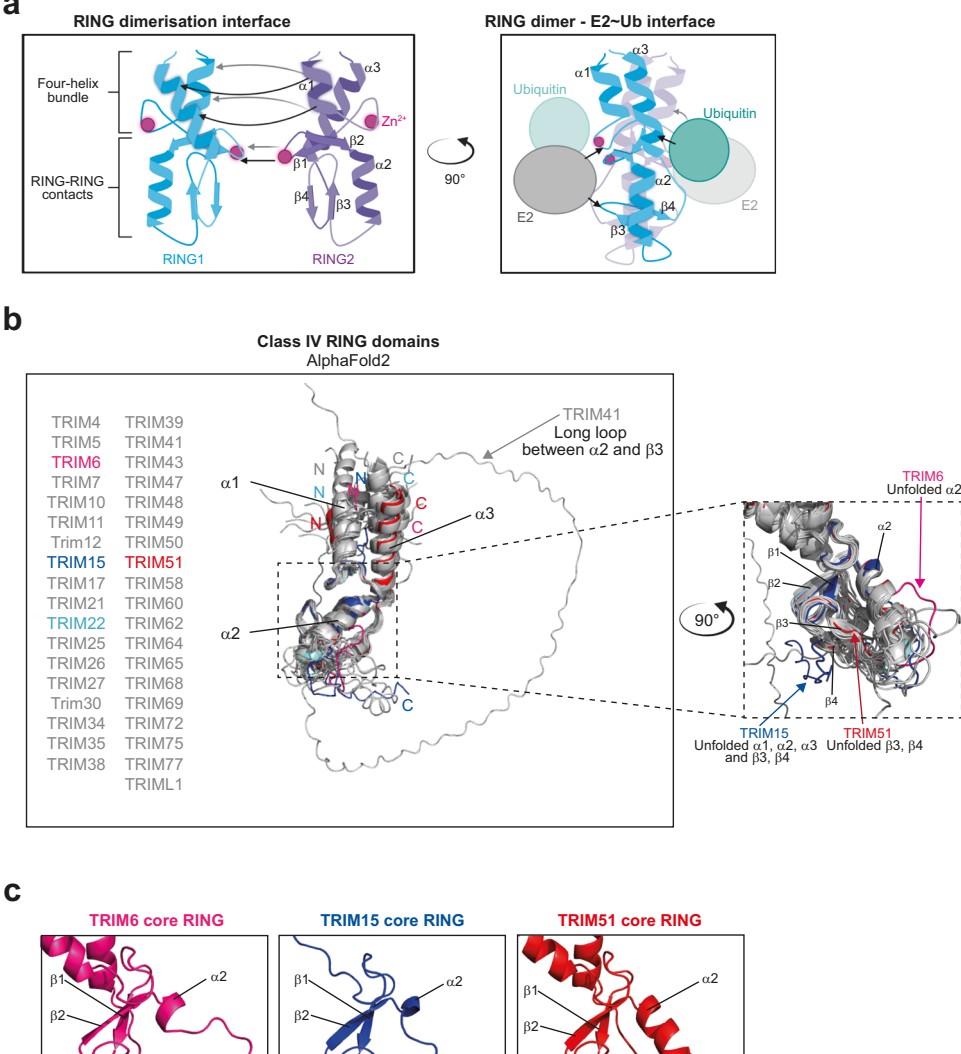

**Fig. 2 | in silico analyses of TRIM family proteins highlights notable structural differences in the RING domain. a** Schematic representations of binding interfaces between TRIM RING homodimers (left) and between TRIM RING dimers and the E2-ubiquitin conjugate (right). Created in BioRender. Dudley, J. (2025) https://BioRender.com/v35y632. **b** Aligned AlphaFold2 structural predictions of the RING domains of class IV TRIMs (N-terminal Met to end of α3 helix, or with a sequence of a corresponding length where no α3 helix is predicted), with divergent structural features of TRIMs 6, 15 and 51 highlighted in an enhanced zoom of the core RING domain. **c** AlphaFold2 predictions of core RING domains of TRIMs 6, 15 and 51.

hydrophobic core via contacts with β3-β4; TRIM15 is predicted to be largely unstructured and lack the dimerisation-mediating α1 and α3 helices; and TRIM51 is predicted to have an unfolded β3-β4 region in the core RING domain (Fig. 2c). We then used these models to guide biochemical analysis of the impact of these predicted RING structural variants on ubiquitin E3 ligase activity.

## TRIM15 RING domain is a monomer that does not show ubiquitin E3 ligase activity

TRIM15 (class IV) is predicted to contain a RING domain according to sequence-based alignments and the presence of zinc co-ordinating residues, yet AlphaFold2 predicts that TRIM15 lacks the α1 and α3 dimerisation-mediating helices, likely due to a high proportion of helix-disrupting proline and glycine residues in these regions, and other secondary structural features generally present in RING domains (Fig. 3a, b). Inclusion of $Zn^{2+}$ and modelling of a dimeric RING arrangement using AlphaFold3 predicts a better-folded α2, while RoseTTAFold2 predicts a better-folded β3-β4 region, yet none of the models predict α1 and α3 to be folded. To assess the accuracy of these predictions, we interrogated the potential self-association of the TRIM15 RING using SEC-MALLS, as RING dimerisation requires properly folded α1 and α3. This analysis showed that even at high concentrations TRIM15 RING remains monomeric (Fig. 3c). Moreover, the 2D $^1$H-$^{15}$N HSQC NMR spectrum of TRIM15 RING, at 1 mM concentration, exhibits sharp cross peaks with a signature similar to that of homogeneous small proteins (Fig. 3d). The spectrum displays features characteristic of a folded protein with a subset of resonances outside the random-coil region. This suggests, in agreement with the RoseTTAFold2 prediction, that TRIM15 RING might possess additional elements of secondary structure. To test whether TRIM15 may function with different E2 enzymes despite its monomeric status, further ubiquitination assays were carried out with either full-length TRIM15 protein isolated from HEK293T cells or recombinantly purified TRIM15 RING domain using a wide range of E2s, including UBE2N/V2, which was previously seen to function with TRIM15 (Fig. 3e, Supplementary Fig. 4a). However, TRIM15 was unable to catalyse ubiquitin chain formation in these assays or to discharge ubiquitin from an E2-ubiquitin$^{ATTO}$ thioester conjugate onto free lysine in solution, in contrast to the positive controls TRIM2 or TRIM21 (Fig. 3f, Supplementary Fig. 4b). Taken together, these data suggest TRIM15 is another example of a TRIM with a monomeric RING domain that shows no apparent ubiquitin ligase activity, similar to TRIM3[17]. Previously, we showed that TRIM3 E3 ubiquitin ligase activity can be rescued by forcing its dimerisation in a tandem RING construct[17]. To test whether TRIM15 activity could be similarly rescued, two RING domains (residues 2–80) were expressed in tandem and ubiquitin discharge and ubiquitin chain formation were analysed (Supplementary Fig. 4c–e). However, E3 ubiquitin ligase activity remained undetectable alongside a positive control, TRIM21. Furthermore, AlphaFold3 modelling of the TRIM15 tandem RING construct offers no consensus orientation, suggesting that the two RINGs cannot be directed into a stable conformation that is conducive to ligase activity, (Supplementary Fig. 4f). Taken together, these data indicate that the RING domain does not exhibit any propensity for self-association or ubiquitination activity, indicating that TRIM15 may either require additional factors for activity, or may carry out alternative non-ubiquitin E3 ligase-related roles in cells. A more detailed investigation of TRIM15 would be of interest in this regard as previous studies have outlined roles for both ubiquitin ligase-dependent pro-tumourigenic cell signalling as well as RING-independent regulation of focal adhesions and inflammatory innate immune signalling[36–38].

## Key E2-ubiquitin contacts are mutated in TRIM6 and TRIM22, leading to lack of ubiquitin ligase activity in vitro

TRIM6 and TRIM22 similarly did not demonstrate ubiquitin ligase activity in in vitro auto-ubiquitination assays detected by ELISA

(Fig. 2b) or western blotting (Supplementary Fig. 5a) using protein isolated from mammalian cells. Furthermore, recombinantly purified TRIM6 RING and RBCC constructs did not auto-ubiquitinate in vitro with previously reported cellular E2 partners (Supplementary Fig. 5b–d)[39,40]. As previous studies have shown that other ubiquitination-deficient TRIMs are inactive because of their inability to dimerise, we tested whether this was the case for TRIMs 6 and 22. However, SEC-MALLS data show that recombinant TRIM6 and TRIM22 RING domains form dimers in solution, with TRIM22 existing in a monomer-dimer equilibrium under these experimental conditions (Fig. 4a).

TRIM6 and TRIM22 have both been reported to drive innate immune responses to viral infection. Their ubiquitin E3 ligase activity has been demonstrated following infection or interferon signalling, which generated either free ubiquitin chains that activate innate immune IKKε signalling or ubiquitination and degradation of viral substrates[39,41,42]. To test whether activation of interferon (IFN) signalling may activate the E3 ligase function of TRIMs 6 and 22, cells over-expressing FLAG-tagged TRIM6, TRIM22, or TRIM2 (an active TRIM control) were treated with IFN-β or TNF-α for 18 h before FLAG immunoprecipitation followed by an in vitro ubiquitination assay. Encouragingly, we found that TRIM6- and TRIM22-mediated ubiquitination could be strongly induced by IFN-β (Supplementary Fig. 5e, f). To understand the factors that could be inducing E3 ligase activity in TRIMs 6 and 22, we performed global proteome and interactome mass spectrometry analyses. These data reveal that IFN-β treatment strongly upregulated the expression of several TRIMs, including TRIM21, as has been reported elsewhere[43], which co-immunoprecipitated with TRIM6 and 22 (Supplementary Fig. 5g, h). However, as TRIM21 can function as an intracellular antibody receptor that recognises antibody-coated immune complexes and induces their degradation[13], we questioned whether IFN-β-induced TRIM21 binds the antibody used for this immunoprecipitation rather than interacting directly with TRIM6 or 22 and therefore might be responsible for the ubiquitination activity observed, independent of TRIM6 or TRIM22. Indeed, all IFN-β-responsive activity of TRIMs 6 and 22 was lost when GFP-tagged TRIMs were purified using antibody-free 'GFP clamp' DARPin pull down instead of immunoprecipitation (Supplementary Fig. 5i). Together, these data show that TRIM21 has the capacity to confound studies of ubiquitination that employ antibody affinity methodologies in IFN-driven signalling contexts, hence the use of alternative techniques is imperative.

In light of these results, we revisited AlphaFold structural predictions to better understand the properties of the RING domains of TRIM6 and TRIM22. Strikingly, the models suggest that α2 in the TRIM6 core RING is likely to be unfolded and several hydrophilic residues are predicted to lie in the α2 to β3-β4 interface in both TRIMs 6 and 22. This contrasts with the well-folded core found in active TRIM class IV members (Figs. 2 and 4b, c). Moreover, the ends of α1 and α3 in TRIM22 are predicted with low confidence (pLDDT < 75), potentially due to two disruptive Lys at the interface of α1 and α3 (Fig. 4b, Supplementary Fig. 6a). To probe these predictions experimentally, we recorded 2D $^1$H-$^{15}$N HSQC of the RINGs of TRIM6 and TRIM22 (Supplementary Fig. 6b). While the TRIM6 RING spectrum shows sharp well-dispersed cross peaks indicative of a single folded species, that of TRIM22 RING displays a degree of line broadening with the majority of sharp cross peaks clustered in the central area of the spectrum. This might indicate that different monomer-dimer equilibria occur at different NMR time scales for the two RING domains and may involve, in the case of TRIM22 RING, an intermediate folding of α1 and α3 upon dimerisation. Finally, sequence alignment revealed that TRIMs 6 and 22 also both have a Gln in place of the key positively charged His, Lys, or Arg 'linchpin' residue observed in the majority of RING domains. Although some TRIM RING domains with E3 ligase activity have alternative residues in this position (e.g. Ser in TRIM32, PDB: 5FEY), in the

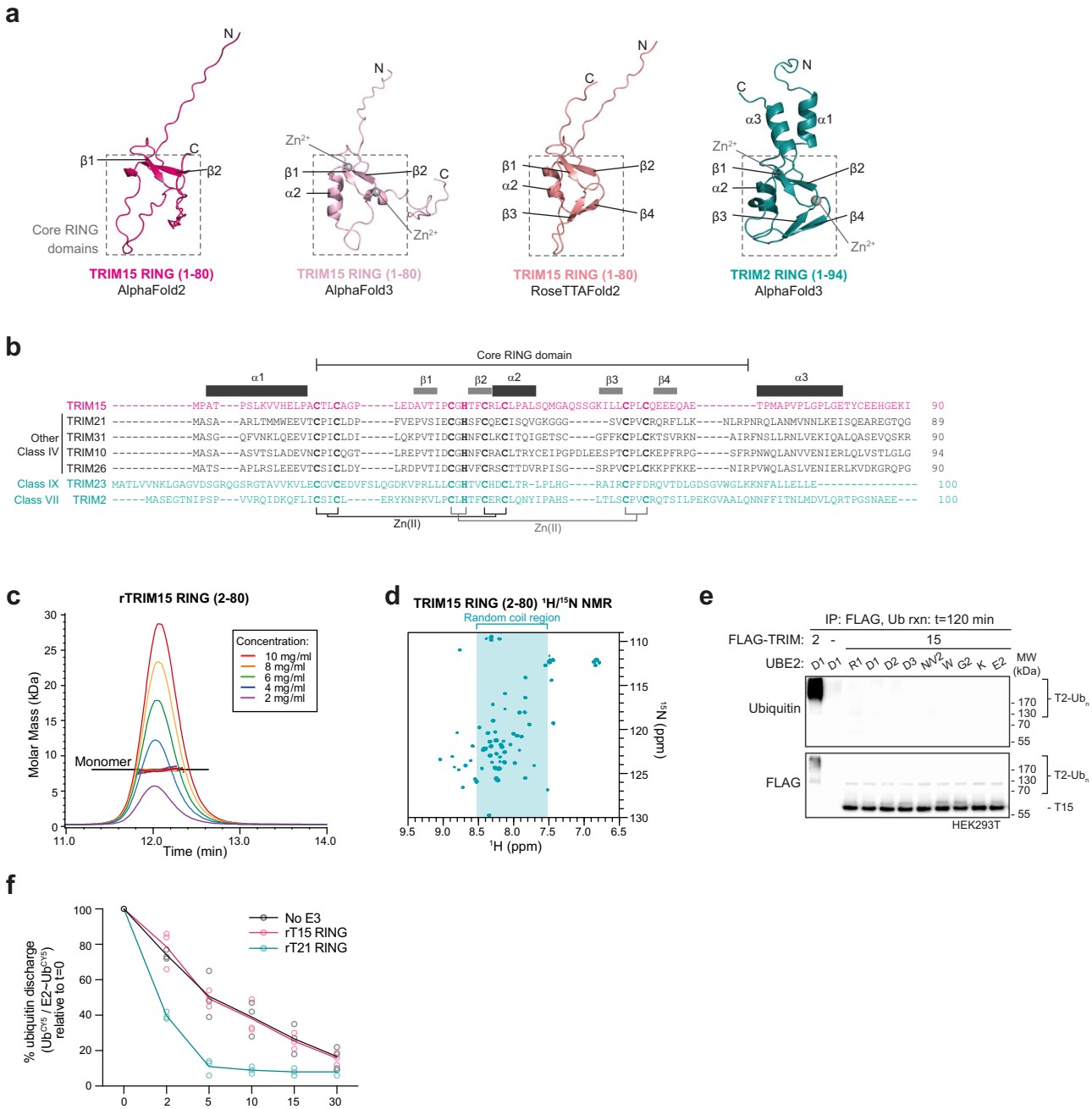

**Fig. 3 | TRIM15 has a monomeric RING that does not show ubiquitin ligase activity in isolation. a** Structural prediction of TRIM15 RING domain modelled by AlphaFold2 (hot pink, left), AlphaFold3 (predicted as a dimer with $Zn^{2+}$ (grey), only one monomer shown, light pink, middle left), RoseTTAFold2 (salmon pink, middle right) and the TRIM2 RING domain structure predicted by AlphaFold3 (predicted as a dimer with $Zn^{2+}$ (grey), one monomer shown, teal, right), with grey dashed lines denoting the core RING domain region. **b** Annotated sequence alignment of TRIM15 with active TRIMs in the same class (black) or different classes (teal). **c** SEC-MALLS analysis of TRIM15 RING domain (residues 2–80) analysed at multiple concentrations. **d** $^{1}H$-$^{15}N$ 2D HSQC spectrum of TRIM15 RING domain, with the spectral region corresponding to resonances characteristic of residues in random-coil conformation indicated in teal. **e** Western blot analysis of auto-ubiquitination reaction carried out using 1 μM of each of the indicated E2 enzymes (UBE2R1, D1, D2, D3, N/V2, W, G2, K or E2), using FLAG pull downs from either untransfected, FLAG-TRIM2, or FLAG-TRIM15 transfected HEK293T cells (n = 3 independent experiments). **f** Quantification of the discharge of $Ub^{CY5}$ from pre-charged UBE2D1 onto free lysine in solution, with or without 4 μM recombinant TRIM21 or TRIM15 RING domains (see Supplementary Fig. 4b) (n = 3 independent experiments). Circles: individual values, error bars: mean ± SEM. Source data are provided as a Source Data file.

previous structures of TRIM RING domains bound to E2-Ub conjugates (PDB 7ZJ3, 7BBD 6S53, 5VZW, 5FER, 5EYA) the Arg linchpin in the RING domain interacts with both the E2 and Ub molecules in E2-Ub conjugate to promote a 'closed' conformation; an interaction that is stabilised by a hydrophobic core between α2 and β3-β4 of the RING domain. Interestingly, the binding mode of the E2-Ub conjugate with

active RING domains lacking a positively charged residue in this position is currently unknown. Furthermore, in addition to the proximal Ub, the closed E2 conformation also binds a 'backside' ubiquitin molecule, which is required for the efficient discharge of ubiquitin[44].

To assess the importance of these features, mutant constructs of TRIM6 and TRIM22 were generated to correct: a) the disrupting

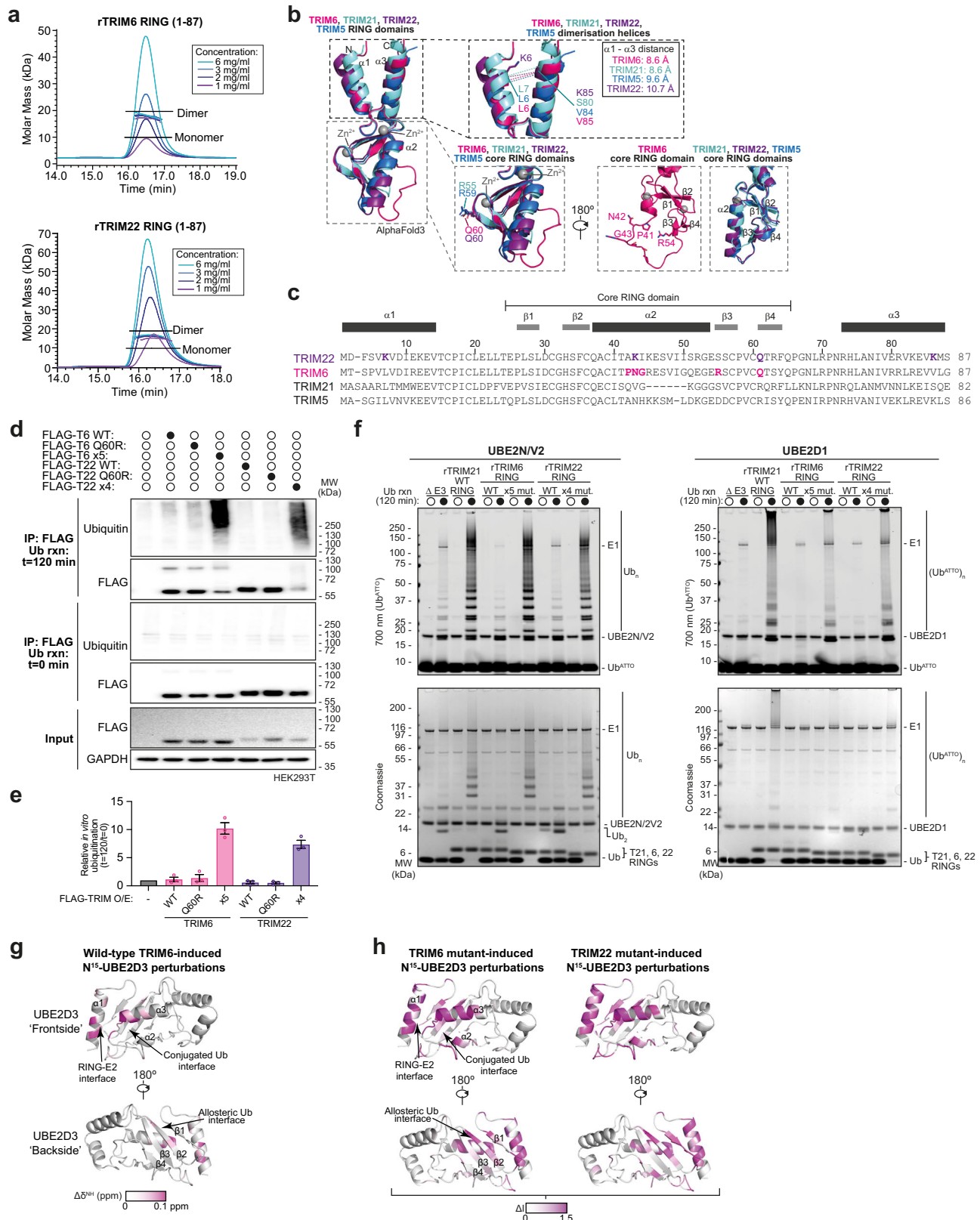

residues in the α2 helix (TRIM6 P41A/N42V/G43I), b) the RING's hydrophobic core (TRIM6 R54S and TRIM22 K42V), c) the lysines in α1 and α3 (TRIM22 K6L/K85V) and d) the linchpin (TRIM6 Q60R and TRIM22 Q60R). Strikingly, TRIM6 penta- and TRIM22 tetra-mutants elicit strong ubiquitin ligase activity in both full-length over-expressed proteins immunoprecipitated from mammalian cells and

RING domains in isolation recombinantly purified from bacteria (Fig. 4d–f, Supplementary Fig. 6c). TRIM6 and TRIM22 mutants functioned with UBE2D family members and the dimeric E2 UBE2N/ 2V2 at comparable levels to closely related class IV member TRIM21. Consequently, we hypothesised that wild-type TRIM6 and TRIM22 are inactive due to linchpin-deficient, destabilised RING domain

**Fig. 4 | Key interfaces with E2-ubiquitin are mutated in TRIM6 and TRIM22, leading to lack of ubiquitin ligase activity in vitro. a** SEC-MALLS analysis of TRIM6 an TRIM22 RING domains (residues 1–87) analysed at multiple concentrations. **b** Aligned Alphafold3 structural predictions of the closely related class IV TRIMs 5, 6, 21 and 22 (predicted as dimers with $Zn^{2+}$ co-ordination, monomers shown). Black dashed lines outline the α1 and α3 helices involved in RING dimerisation, grey dashed lines denote the core RING domain region. Relevant amino acid residues are labelled. **c** Protein sequence alignment of TRIMs 5, 6, 21 and 22, with relevant residues highlighted in bold. **d** Western blot analysis of auto-ubiquitination reactions carried out using 1 μM UBE2D1 and FLAG pull downs from HEK293T cells expressing FLAG-tagged full-length wild-type (WT), linchpin mutant (Q60R), or active mutant (TRIM6 ×5, P41A/N42V/G43I/R54S/Q60R; TRIM22 ×4, K6L/K42V/Q60R/K85V) TRIM6 or TRIM22 (n = 3). **e** Bar graph representing the quantification of western blots of the experiment shown in (**d**). Circles: individual values, error bars: mean ± SEM, n = 3. **f** 700 nm fluorescent imaging ($Ub^{ATTO}$) and Coomassie stains of SDS-PAGE gels of auto-ubiquitination reactions carried out using 2 μM UBE2N and UBE2V2 (left) or 2 μM UBE2D1 (right), with 1 μM UBA1, 50 μM ubiquitin, 1 μM $Ub^{ATTO}$ and 3 mM ATP, using recombinantly purified RING wild-type (WT) or active mutants (TRIM6 ×5, P41A/N42V/G43I/R54S/Q60R; TRIM22 ×4, K6L/K42V/Q60R/K85V) TRIM6 or TRIM22 (n = 3). **g** $^{1}$H/$^{15}$N chemical shift perturbations ($\Delta\delta^{NH}$, pink gradient) induced by the presence of wild-type TRIM6 RING domain mapped onto the structure of UBE2D3 (5EGG, grey). **h** As in (**g**), except regions of $^{15}$N UBE2D3 (5EGG, grey) interaction with ×5 mutant TRIM6 (left) or ×4 mutant TRIM22 (right) are represented by degree of line broadening ($\Delta I$, pink gradient). Source data are provided as a Source Data file.

cores that on their own are unable to promote an active 'closed' E2-Ub conjugate.

To test this model, we used NMR to monitor the changes in the $^{1}$H-$^{15}$N HSQC spectra of $^{15}$N-labelled UBE2D3 at increasing concentrations of unlabelled TRIM6 and TRIM22 RING. Surprisingly, fast exchange was observed in an equivalent set of cross peaks in both titration experiments. The mapping of the observed small chemical shift perturbations (CSPs) on the available UBE2D3 crystal structure (PDB: 5EGG) shows an interface encompassing part of helix α1, the loop region between α2-α3 and the N-terminal side of helix α3 (Fig. 4g, Supplementary Fig. 6d). The E2 resonances affected in our experiments are only a subset of those previously reported in the interaction of UBE2D3 with the RING domain of the active ubiquitin E3 ligase TRIM2[17]. The interaction of TRIM2 RING with the E2 produced stronger CSPs that extended to a number of residues further away from the E2/RING interface, encompassing the whole α2-α3 region, which accommodates the conjugated ubiquitin in closed conformation, as well as the allosteric ubiquitin-binding site that surrounds residue S22 on the E2 backside. These areas are unperturbed in the interaction with both TRIM6 and TRIM22 RINGs (Fig. 4g).

To assess whether mutations of TRIM6 and TRIM22 that restore E3 activity were a consequence of their impact on E2 binding, we titrated a $^{15}$N-labelled sample of UBE2D3 with TRIM6 and TRIM22 mutant forms. We observed, at increasing concentration of both TRIM RING ligands, a gradual disappearance of the majority of the resonances in the E2 spectrum to a level below detection, especially apparent for TRIM22 (Supplementary Fig. 6f, g). The line broadening effect is consistent with the formation of a dimeric E2-RING complex, with higher molecular weight (~55 kDa), invisible in the $^{1}$H-$^{15}$N HSQC spectrum due to its longer correlation times, in equilibrium with the free species. The resonance broadening pattern has the same signature for both TRIM6 and TRIM22 RING mutant forms. Mapping the affected residues on the E2 structure highlighted a large part of the protein affected by the binding. As in the case of TRIM2, the strongly affected residues are directly in the RING/E2 interface, on the surface involved in the stabilisation of the conjugated ubiquitin in closed conformation and the backside allosteric ubiquitin interface[17]. Interestingly, the TRIM22 mutant appears to have a more pronounced effect on the backside residues than TRIM6, which correlates with their activity patterns observed with UBE2D3 (Supplementary Fig. 6c).

Taken together, our analyses suggest that TRIM6 and TRIM22 have structural and sequence features that are not compatible with canonical TRIM RING ubiquitin ligase catalytic requirements, although the possibility that these proteins may function under specific reaction conditions cannot be ruled out. Consequently, there is a need for future studies to understand the apparent inconsistency between the in vitro data described here and previous cell-based studies reporting a role for TRIM6 and TRIM22 ubiquitin ligase activity in innate immune signalling[39–41,45].

## TRIM49 forms a regulatory interaction with its inactive paralogue, TRIM51, with implications in autophagy

TRIM51 is a paralogue of TRIM49 (both class IV), sharing 75% sequence identity overall, with 73% in their RING domains and 76% in the RBCC motif. However, TRIM51 did not demonstrate E3 ligase activity in vitro, whereas TRIM49 did (Fig. 1b). Structural predictions suggest that the β3-β4 region in the core of TRIM51's RING domain may not be well-folded, with positively charged (Lys and Arg) and small (Ala and Val) residues in place of a negatively charged Glu and bulky hydrophobic residues (Pro and Phe) in TRIM49 (Figs. 2 and 5a, b). We therefore hypothesised that TRIM51 lacks the hydrophobic core found in structurally characterised RING domains of active TRIMs (e.g. TRIM25 RING, PDB: 5FER). Consequently, the RING domain of TRIM51 is unlikely to enable a stable interaction with the E2-Ub conjugate, similarly to TRIMs 6 and 22 (Fig. 4). Consistent with this model, swapping the β3-β4 region from TRIM51 into TRIM49 rendered it inactive (Fig. 5c, d). Unfortunately, attempts to recombinantly express and purify TRIM49 and TRIM51 RING domains to directly assess their interaction with a cognate E2 were unsuccessful due to solubility issues (data not shown).

Next, we questioned whether, similarly to several recent reports of homologous TRIM 'pairs', TRIM49 and TRIM51 may exhibit a functional interaction with each other[17,46,47]. Interestingly, we find that TRIM49 and TRIM51 interact via their coiled-coil domains and co-localise to punctate structures in cells (Fig. 5e–h, Supplementary Fig. 7). Moreover, we find that TRIM51 co-overexpression represses TRIM49 ubiquitin ligase activity, suggesting a regulatory function for TRIM51 (Fig. 5e, f).

We then wanted to understand the functional significance of the regulation of TRIM49. Previously, autophagic flux screens have identified TRIMs 49 and 51 as potential positive and negative autophagy regulators, respectively[24,48], and, in line with their opposing ubiquitination activities, we similarly observe that overexpression of the active ubiquitin ligase TRIM49 increases both basal and amino acid starvation-induced autophagic flux, whereas TRIM51 overexpression reduces LC3B lipidation (Fig. 5i, j). Moreover, TRIM49 co-localises with LC3B-positive puncta during both untreated and amino acid-deprived conditions, whereas TRIM51 exhibits diffuse staining in irrespective of nutrient levels (Fig. 5k, l). When overexpressed with TRIM49, however, TRIM51 co-localises to TRIM49-positive spots, suggesting that TRIM51 can be recruited to TRIM49-positive puncta (Fig. 5g).

## Discussion

The TRIM family of proteins are implicated across a fascinating spectrum of biology, from immunity to neurodevelopment and tumourigenesis[4,49]. However, the biochemical mechanisms underlying these observed cellular roles remain largely unexplored for most family members. Our parallel analyses of structure-function relationships of RING domains across the whole TRIM protein family have revealed previously unappreciated ubiquitin E3 ligase-defective TRIMs, which we term 'pseudoligases'.

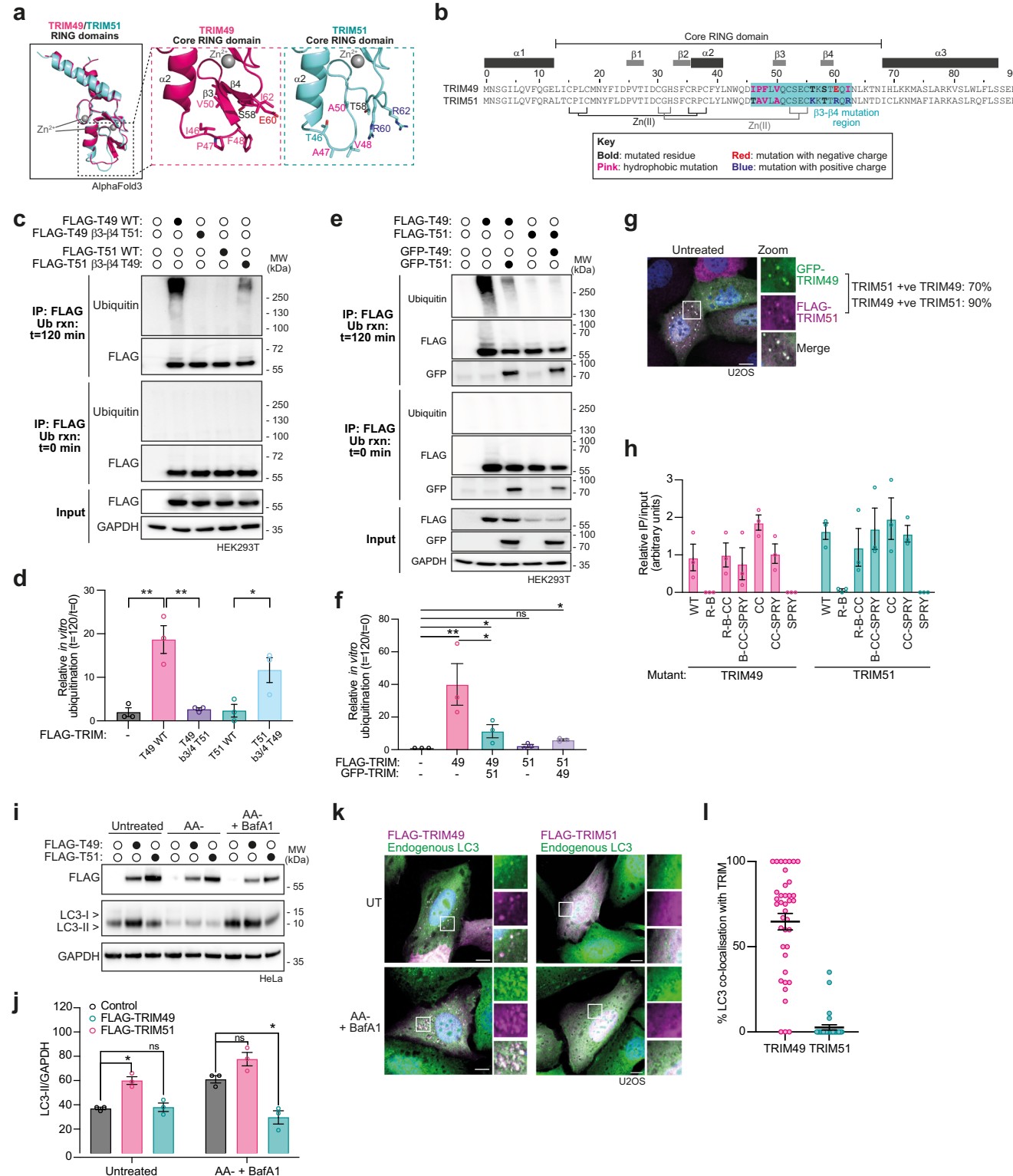

The concept of 'pseudoenzymes' has been well described elsewhere, particularly in the kinase field, where evolution has generated branches of structural divergence that spawn novel functions that often involve regulation of active kinases (e.g. activation or inhibition as part of heteromeric complexes)[50]. Here we demonstrate that the RING domains of TRIMs 6, 15, 22 and, most likely, 51 do not catalyse canonical ubiquitination and, given their structural properties, are unlikely to do so under differing experimental conditions. This led us to define these proteins as pseudoligases and suggests that they have

alternative, potentially regulatory, roles and may, for example, be involved in the pairing of active and inactive RING domains, as is well described for other RING E3s such as BRCA1 and BARD1[51]. In light of the reported importance of the identified pseudoligase TRIMs in cancer and innate immunity, there is now an evident need to contextualise the data presented here and clarify the cellular functions of these proteins[41,45,52–56].

For the majority of active TRIMs, it is largely unknown whether substrate ubiquitination in cells is constant or triggered in response to

**Fig. 5 | TRIM49 and TRIM51 form an active:inactive pair that cross-regulate, with conflicting effects on autophagy. a** AlphaFold3 structural predictions of RING domains of TRIM49 (pink) and TRIM51 (aqua) (predicted as dimers with $Zn^{2+}$, monomers shown). **b** Sequence alignment of TRIM49 and TRIM51. **c** Western blots of FLAG-tagged wild-type (WT) or mutant TRIM49 and 51 proteins immunoprecipitated from HEK293T cells, which were then used in an in vitro ubiquitination assay with 1 μM UBE2D1 (n = 3). **d** Quantification of n = 3 independent experiments as described in (**c**). Circles: individual values, error bars: mean ± SEM (two-tailed *t*-tests, *P* values left to right: 0.0075, 0.0075 and 0.0433 (ns > 0.05, * < 0.05, ** < 0.01, *** < 0.001)). **e** Western blots of FLAG-tagged TRIM proteins immunoprecipitated from HEK293T cells, with or without co-expression of GFP-tagged TRIMs, that were used in an in vitro ubiquitination assay as described for (**c**) (n = 3). **f** Quantification of n = 3 independent experiments as described in (**e**). Circles: individual values, error bars: mean ± SEM (two-tailed *t*-tests, *P* values left to right: 0.0030, 0.2254, 0.0375, 0.0072 and 0.0268 (ns > 0.05, * < 0.05, ** < 0.01, *** < 0.001)). **g** Representative images showing the co-localisation of GFP-TRIM49 (green) with

FLAG-TRIM51 (magenta) in U2OS cells, with DAPI-stained nuclei in blue (n = 3). Scale bars: 10 μm. **h** Quantification of western blotting of n = 3 independent experiments analysing the interaction of truncation mutants FLAG-TRIM49 and GFP-TRIM51 using co-immunoprecipitation (n = 3). Circles: individual values, error bars: mean ± SEM (Supplementary Fig. 7). **i** Western blots against whole cell lysates of HeLa cells either untreated or treated for 4 h HBSS (AA−) ± 50 nM Bafilomycin A1 (BafA1) (n = 3). **j** Quantification of western blots of LC3-II relative to GAPDH (relating to **i**) (n = 3). Circles: individual values, error bars: mean ± SEM (two-tailed *t*-tests, *P* values left to right: 0.0357, 0.7868, 0.1862 and 0.0498 (ns > 0.05, *<0.05, **<0.01, ***<0.001)). **k** Representative images showing the co-localisation of FLAG-TRIM49 or -TRIM51 (magenta) with endogenous LC3 (green) in U2OS cells (n = 3). Scale bars: 10 μm. **l** Quantification of percentage (%) of LC3 puncta that co-localise with either TRIM49 or TRIM51. Circles represent the proportion of LC3/TRIM co-localisation per cell (n = 3 independent experiments), error bars: mean ± SEM. Source data are provided as a Source Data file.

specific stimuli, although diverse transcriptional, localisation and phosphorylation mechanisms have been proposed in some cases[43,57,58]. Additionally, it is becoming increasingly appreciated that TRIMs can regulate each other's activity in *trans*: TRIM2 and TRIM3 (67% identity) interact in neuronal cells and impact each other's activities; TRIM9 and TRIM67 (62% identity) interact and also compete for a substrate, with opposing impacts on its ubiquitination; and TRIM17 and TRIM41 (35% identity) interact such that TRIM17 represses the ubiquitin ligase activity of TRIM41[17,46,47]. We were therefore intrigued to find that pseudoligase TRIM51 adds to this expanding picture of 'pairs' of closely related TRIMs through its interaction with ligase-proficient TRIM49 (75% identity), which corresponds to their opposing effects on autophagic flux. Further research is required to understand whether the formation of hetero-oligomeric TRIM complexes is a more universal common evolutionary mechanism to modulate TRIM ubiquitin E3 ligase activity.

As part of this study, we carried out both in cellulo and in vitro ubiquitination analyses across the whole TRIM family, which has offered a unique insight into the advantages and assumptions inherent in these approaches. Much of the research regarding TRIM proteins has relied on assessments of protein ubiquitination in cells using overexpressed tagged ubiquitin (Supplementary Table 2). Whilst this approach has the advantage that potentially unappreciated binding or regulatory partners are present, as well as a variety of cognate E2 enzymes, it assumes that the putative ligase of interest is directly ubiquitinating the substrate in question (or auto-ubiquitinating itself). However, it may be the case that the putative ligase in fact acts upstream, or functions as a recruiting factor, and another ligase instead ubiquitinates the substrate. This is particularly auspicious in the case of TRIMs that have been ascribed alternative functions (e.g. SUMOylation) and pseudoligase TRIMs[9,10,59,60]. Whilst the ability to produce recombinant proteins can be a limiting factor in some instances, complementing in cellulo ubiquitination with in vitro analyses offers important information regarding bona fide enzymatic functions. However, both in cellulo and in vitro assays presented here are semi-quantitative methods that cannot assess absolute levels of enzymatic processivity of each TRIM and, in the case of the ELISA assay, whether a high signal infers many short ubiquitin chains or fewer longer chains. Additionally, these methods cannot exclude the possibility that these TRIMs may act as E3 ligases for ubiquitin-like modifiers or could ubiquitinate non-proteinaceous substrates[20]. Moreover, we provide a cautionary observation that TRIM21 is a confounding factor in ubiquitination analyses when using antibody pull downs to isolate proteins from cells that have elevated innate immune signalling. Therefore, it is key to employ non-antibody-based methodologies to research ubiquitination research in the context of infection.

Furthermore, the results presented here may inform the efficacy of certain TRIMs as the ubiquitin-conjugating pair in targeted protein

degradation approaches (e.g. PROTACs or molecular glues), given their differing abilities to catalyse ubiquitination. Furthermore, understanding the functional activities employed across the whole TRIM family may uncover novel mechanistic details that, given the well-documented links between TRIM proteins and a multitude of pathologies, could subsequently lead to new therapeutic opportunities[3,4].

Taken together, we anticipate that the findings presented here may lead to many exciting future avenues of research in the field of TRIM biology, both ubiquitin-dependent and -independent.

## Methods

### Cell culture and treatments
HEK293T, HeLa and U2OS cells were cultured in Dulbecco's modified eagle medium (ThermoFisher, 41966-029), supplemented with 10% FBS (ThermoFisher, 10270106) and 100 U/ml penicillin/streptomycin (Gibco, ThermoFisher Scientific), at 5% $CO_2$ at 37 °C.

### Recombinant protein expression and purification
Cloning, expression and purification of the E1 (UBA1) and E2 (UBE2D1, UBE2D2, UBE2D3, UB2E1, UBE2E2, UBE2G2, UBE2K, UBE2N, UBE2V2, UBE2R1, UBE2W) enzymes have been described previously[17,18,61,62]. UBE2D1-Ub[ATTO] and UBE2D1-Ub[CY5] were prepared as described previously[62,63]. Ubiquitin was purchased commercially (Sigma, U6253, from bovine erythrocytes). Human TRIM2 RING-B-box-coiled-coil (RBCC: 2–316); TRIM6 RING (R, 1–87 WT or P41A/N42V/G43I/R54S/Q60R mutant); TRIM22 R (1–87 WT or K6L/K42V/Q60R/K85V mutant); TRIM21 R (1–85); TRIM23 RBB (2–213); TRIM15 R (2–80); TRIM15 tandem RINGs (2 × 2–80 linked by a Ser) and their corresponding mutant constructs were generated as gBlocks (IDT) and ligated by Gibson Assembly into pET49b vector as HRV 3C protease cleavable N-terminal $His_6$-fusion proteins (see Supplementary Table 4 for full plasmid list)[17]. Briefly, the resulting $His_6$-tagged proteins produced from BL21 affinity were purified by Ni-NTA affinity chromatography (ThermoFisher, 88222) in 50 mM HEPES pH 7.5, 300 mM NaCl, 20 mM imidazole and 0.5 mM TCEP. This was followed by further purification by size-exclusion chromatography with Superdex S75 XK16/60 column (Cytiva) that had been pre-equilibrated with 50 mM HEPES pH 7.5, 150 mM NaCl and 0.5 mM TCEP. All proteins were stored in a buffer of 50 mM HEPES pH 7.5, 150 mM NaCl and 0.5 mM TCEP.

### Transient protein expression in mammalian cells and immunoprecipitation
HEK293T cells were grown to 60% confluency before transfection with ptCMV-EGFP-TRIM, pcDNA3.1-FLAG-TRIM and/or pCMV-HA-Ubiquitin plasmids for 4 h in OptiMEM (ThermoFisher, 31985062) and 1 μg/ml PEI (Sigma, 764965), as per the manufacturer's instructions (see Supplementary Table 4 for full plasmid list, including mutant constructs). After 24 h further growth in full medium, cells were lysed in 500 μl lysis

buffer (0.5% IGEPAL, 150 mM NaCl, 50 mM Tris-HCl pH 7.5, 5 mM MgCl$_2$, protease inhibitors (Merck, 4693159001)) and centrifugation 14,500 × $g$, 10 min, 4 °C. Lysates were subjected to end-on rotation at 4 °C with anti-GFP (Roche, 11814460001) or anti-FLAG (Merck, A36797) that was pre-conjugated to Protein A/G beads (Pierce, 88802) for 2 h 4 °C (note: pre-conjugation crosslinked using 20 mM dimethyl pimelimidate (DMP, ThermoFisher) in borate buffer (40 mM boric acid, 40 mM sodium tetraborate decahydrate) 45 min RT). The resulting pull downs were washed twice with lysis buffer and once with 0.5 M NaCl lysis buffer then used in various assays as described below. Alternatively, GFP-TRIM6 and -TRIM22 proteins were also pulled down using 15 µg GFP clamp[64] DARPin conjugated to 10 µl NHS-activated magnetic beads (Pierce, 88826) per sample.

## In-cell auto-ubiquitination activity
GFP-tagged full-length TRIM proteins and HA-tagged ubiquitin were transiently co-overexpressed in HEK293T cells and purified as described above, with the additional step that lysis was preceded by 4 h treatment with 10 µM MG132 (Merck, 4693159001) and 10 µM PR619 (Bio-Techne, 4482/10). The HA-ubiquitination status of the resulting purified TRIMs was then assessed by western blotting (as described below).

## Western blot analysis
Samples in SDS loading buffer (Invitrogen, NP0007) were run alongside the PageRuler Plus molecular weight ladder (Thermo, 26619) on 4–12% gradient NuPAGE Bis-Tris gels (Invitrogen) in MES buffer, then transferred onto nitrocellulose or PVDF membranes (in the case of LC3 blotting) using the TransBlot Turbo system (BioRad). Membranes were blocked in 5% milk in PBS with 0.001% Tween 20 (PBST) before incubation with the specified antibodies prepared in 5% milk (GFP (Roche, 11814460001, 1:1000), FLAG-HRP (Merck, A8592, 1:10,000), HA-HRP (Merck, 3F10, 1:3000), total ubiquitin (Ubi: Invitrogen, 13–1600, 1:1000), conjugated ubiquitin (FK2: Sigma, ST1200, 1:1000; or FK2-HRP: Generon, SMC-214D-HRP, 1:2000), anti-GAPDH (Millipore, MAB347, 1:3000), endogenous LC3 (Sigma, L7543, 1:1000)) which were detected, where required, by anti-mouse-HRP and anti-rabbit-HRP secondary antibodies (either Dako, P0447 and P0399 or Cell Signaling Technologies, 7076 and 7074, all at 1:2000). Finally, detection reagents (Amersham, RPN2106V1&2) were added before imaging with BioRad ChemiDoc and analysis in ImageLab (v6.1.0).

## In vitro auto-ubiquitination activity using mammalian cell-derived protein
For in vitro ubiquitination analyses using FLAG- or GFP-tagged full-length TRIMs isolated from HEK293T cells, reactions of 1 µM E1, 1 µM of indicated E2s (whole-family screen used a mix of 1 µM of each UBE2D1, UBE2N/UBE2V2, UBE2E1, UBE2G2, UBE2W, UBE2K), 50 µM ubiquitin (Sigma, U6253) and 3 mM ATP in reaction buffer (50 mM HEPES pH 7.5, 150 mM NaCl, 20 mM MgCl$_2$) were incubated at 30 °C for the indicated times before either snap freezing (for ELISA analysis) or adding 2X LDS sample buffer (Invitrogen) supplemented with 0.5 M DTT and boiling briefly (for western blotting).

## ELISA to detect in vitro auto-ubiquitination activity
Samples generated by in vitro auto-ubiquitination reactions were separated into technical triplicates and diluted in sample buffer (10 mM sodium phosphate pH 7.2, 140 mM NaCl, 0.05% Tween 20, 1% BSA) to 200 µl final volume per well in Nunc MaxiSorp plates (for optimisation studies, 'low' and 'high' dilutions were 1:5 and 4:5, respectively). Samples were incubated overnight at 4 °C on a rocker. Wells were then washed four times with washing buffer (10 mM sodium phosphate pH 7.2, 140 mM NaCl, 0.05% Tween 20) before adding 250 µl blocking buffer (10 mM sodium phosphate pH 7.2, 140 mM NaCl, 1% BSA) per well for 1 h RT. Wells were washed once with

washing buffer before incubation for 2 h RT with 100 µl anti-ubiquitin-HRP (FK2-HRP: Generon, #SMC-214D-HRP) diluted 1:5000 in blocking buffer. Wells were washed four times with washing buffer before adding 100 µl detection reagents (Amersham, #RPN2106V1&2) and detection of chemiluminescent signal immediately using the CLARIOstar plate reader. To normalise this signal to cellular TRIM expression levels, GFP fluorescence in the whole cell lysates was measured using the CLARIOstar plate reader before processing for the in vitro reaction and ELISA. Note, fatty acid-free (Merck, 10775835001) was substituted for regular (ThermoFisher, 31985947) BSA in the above solutions where indicated.

## in vitro auto-ubiquitination assay using recombinant protein
For in vitro ubiquitination assays recombinant TRIM E3 ligase constructs purified from *E. coli* (as described above) at 4 µM were combined with 1 µM E1, 2.5 µM of each indicated E2, 3 mM ATP, 50 µM ubiquitin (spiked with 1 µM Ub$^{ATTO}$ where indicated) and reaction buffer (50 mM HEPES pH 7.5, 150 mM NaCl and 20 mM MgCl$_2$). Reactions were incubated at 30 °C with agitation for 0, 30, 60 and 120 min, then terminated by the addition of 2x SDS sample buffer (Invitrogen) supplemented with 500 mM DTT and snap frozen. Samples were then analysed by Coomassie staining and detection of in-gel Ub$^{ATTO}$ fluorescence at 700 nm where indicated using an LI-COR CLx scanner.

## Ubiquitin discharge assay
Lysine-discharge assays were carried out at 30 °C using 4 µM recombinant TRIM E3 ligase constructs purified from *E. coli* (as described above) incubated with 1 µM UBE2D1-Ub$^{ATTO}$ pre-charged thioester and 20 mM L-Lysine in reaction buffer (50 mM HEPES pH 7.5 and 150 mM NaCl). Samples were taken at stated timepoints by quenching with 2x SDS sample buffer (Invitrogen) and snap freezing, before resolving by SDS-PAGE and analysis on a LI-COR CLx scanner. Ubiquitin discharge (UBE2D1-Ub$^{ATTO}$/Ub$^{ATTO}$) was quantified by ImageLab Software.

## SEC-MALLS
For size-exclusion chromatography coupled to multi-angle laser light scattering (SEC-MALLS), 120 µl of 0.45 µm filtered protein sample (TRIM15, TRIM6, or TRIM22 RING domains) was used at a range of concentrations (1–6 mg/ml to assess concentration-dependent self-association) and applied to Superdex 200 Increase 10/300 column attached to a Jasco HPLC system. The gel filtration columns were pre-equilibrated in 50 mM HEPES pH 7.5, 150 mM NaCl, 0.5 mM TCEP and 3 mM NaN$_3$, and experimental runs were performed at a flow rate of 0.5 ml/min. Scattering intensity was measured by a DAWN-HELIOS II laser photometer (Wyatt Technology), while the refractive index was measured by an OPTILAB T-rEX differential refractometer (Wyatt Technology). The ASTRA software package (WYATT technology, v7.3.2) was used to determine the average molecular mass and polydispersity across the individual elution peaks.

## NMR
Natural abundance $^1$H-$^{15}$N HSQC spectra of the RING domains of TRIM15 (residues 1–70), TRIM6 (residues 1–87), and TRIM22 (residues 1–87) were acquired at 1 mM protein concentration in 20 mM HEPES pH 7.5, 50 mM NaCl and 0.5 mM TCEP at 298 K on a Bruker AVANCE spectrometer operating at a $^1$H frequency of 700 MHz. The data were recorded using Topspin (Bruker) and processed with NMRPipe[65].

Given the availability of its amide proton chemical shift assignment, NMR experiments were carried out using the UBE2D3 E2 isoform. $^{15}$N-isotope enriched UBE2D3, bearing the mutation of the catalytic cysteine (C85) to serine, was prepared by growing the bacteria in M9 minimal medium using 1 g/L of $^{15}$N-ammonium chloride as sole source of nitrogen. UBE2D3 titrations with wild-type and mutant RING domains of TRIM6 and TRIM22 (residues 1–87, 0 to 2 molar equivalents) were recorded similarly as previously described[18] at 298 K

at a constant concentration of labelled component (150 μM) on a Bruker AVANCE spectrometer operating at a proton frequency of 800 MHz in 25 mM Na-phosphate pH 7.0 and 150 mM NaCl. Data were acquired with Topspin (Bruker), processed with NMRPipe[65] and analysed by CCPNMR[66]. Backbone amide proton and nitrogen nuclei chemical shifts perturbations ($\Delta\delta^{NH}$) observed for the E2 interaction with TRIM6 and TRIM22 RING wild type were calculated using CCPNMR analysis (v2.4.2)[66] and mapped according to their value on the available E2 crystal structure (5EGG). The interaction of the mutant forms of TRIM6 and TRIM22 RING with the E2 has a different NMR signature, with cross peaks in $^1$H-$^{15}$N HSQC gradually broadened with the increasing RING concentration, leaving behind a small number of resonances with unchanged chemical shifts originating from side chains and flexible regions amide protons. To profile the effect of the binding of the RING domain to the E2, we reported the degree of line broadening on the $^1$H-$^{15}$N HSQC spectrum of the E2 at substoichiometric concentration (100 μM) of RING. At this stage no cross peak had been completely 'bleached' as a result of complex formation. We used the quantity $\Delta I = (h0, i/H0) - (hi/Hi)$, where h0, i and hi represent the heights of each resonance from the labelled protein spectrum in the absence and presence of the binding partner, respectively, and H0 and Hi represent the average cross peak heights in the corresponding 2D HSQC NMR spectrum[67]. A non-uniform profile of the quantity $\Delta I$ across the protein chain would indicate a differential line-broadening effect, with larger positive values of $\Delta I$ highlighting regions of the labelled E2 that are most strongly perturbed in the binding. Since proteins are prepared in the same NMR buffer, any change in the spectrum of the labelled E2 in all titrations can be attributed directly to intermolecular interactions. All data were plotted using GraphPad (v10.2.3) and all molecular graphics were prepared using Pymol (v2.5.4).

## Autophagic flux assessment

HeLa cells were grown to 60% confluency in 6-well plates before transfection with 500 ng pcDNA3.1-FLAG-TRIM plasmids for 4 h in OptiMEM (ThermoFisher, 31985062) and 1.5 μl 1 μg/μl PEI (Sigma, 764965) as per the manufacturer's instructions. After 24 h further growth in full medium, ~90% confluent cells were either treated 4 h with HBSS media (ThermoFisher, 24020117) with or without 50 μM Bafilomycin (Merck, B1793), or left untreated. Cells were then lysed in 30 μl lysis buffer (0.5% IGEPAL, 150 mM NaCl, 50 mM Tris-HCl pH 7.5, 5 mM $MgCl_2$, protease inhibitors (Merck, 4693159001)) before centrifugation 14,500 × $g$, 10 min, 4 °C. Lysates were then assessed by western blotting as described above.

## Widefield and confocal microscopy

To screen for TRIM localisation, U2OS cells were cultured to 50% confluency on 22 mm coverslips (VWR, 631-1582) in 6-well plates before transfection with 500 ng ptCMV-EGFP-TRIM plasmids for 4 h in OptiMEM (ThermoFisher, 31985062) and 1.5 μl 1 μg/μl PEI (Sigma, 764965) as per the manufacturer's instructions. After 24 h further growth in full medium, cells were fixed with 4% paraformaldehyde (ThermoFisher, 15670799) 30 min RT, permeabilised with 0.1% Triton X-100 (Sigma) in PBS 5 min RT and stained with 1 μg/ml DAPI (Merck, D9542) in PBS for 10 min before mounting on slides with ProLong Gold Mountant (ThermoFisher, P10144). Slides were imaged on a Nikon Long-Term Time-Lapse (LTTL) widefield microscope, with at least n = 3 per TRIM.

To assess TRIM49 and TRIM51 co-localisation with each other or endogenous LC3, U2OS cells were cultured to 50% confluency on 22 mm coverslips (VWR, 631-1582) in 6-well plates before either transfection with 500 ng pcDNA3.1-FLAG- or ptCMV-EGFP-TRIM49 or -TRIM51, plasmids, as specified, for 4 h in OptiMEM (ThermoFisher, 31985062) and 1.5 μl 1 μg/μl PEI (Sigma, 764965) as per the manufacturer's instructions. After 16 h further growth in full medium, cells were either left untreated or treated 4 h with HBSS media (ThermoFisher, 24020117) supplemented with 100 μM Bafilomycin (Merck, B1793). Cells were then fixed with 4% paraformaldehyde (ThermoFisher, 15670799) 30 min RT, permeabilization with 0.1% Triton X-100 (Sigma) in PBS 5 min RT, blocked with 1% BSA PBS, stained with 1:100 anti-FLAG (Merck, F1804) or 1:50 anti-LC3 (Sigma, L7543) in 1% BSA PBS O/N 4 °C, then 1:200 anti-mouse-Alexa594 (ThermoFisher, A11032) or anti-rabbit-Alexa-488 (ThermoFisher, A11008) 1 h, before counterstaining with 1 μg/ml DAPI (Merck, D9542) in PBS for 10 min, and finally mounting on slides with ProLong Gold Mountant (ThermoFisher, P10144). Slides were imaged with 63× magnification oil immersion lens on Zeiss Invert880 confocal microscope, taking 12 × 0.36 μm z-slices per field of view, with at least n = 3 per sample. Images were analysed in ImageJ software (v2.16.0).

## Mass spectrometry global proteomic and interactome analyses

For TRIM6 and TRIM22 mass spectrometry analyses, biological triplicates of 4 × 10⁶ HEK293T cells transfected with pcDNA3.1-FLAG empty vector (control), pcDNA3.1-FLAG-TRIM6, or pcDNA3.1-FLAG-TRIM22, either untreated or treated 18 h 5000 units/ml IFN-β (R&D Systems, #11410-2), were lysed in 2 ml lysis buffer (0.5% IGEPAL, 100 mM KCl, 50 mM Tris pH 7.5, 5 mM $MgCl_2$, with protease inhibitors (Merck, 4693159001)). 2.5% of the sample was taken for global proteomics, to which equal volume 5% SDS, 100 mM TEAB was added and samples were incubated 37 °C, 5 min. Samples were processed using S-trap (Protifi) micro protocol according to the manufacturer's instructions. For interactome analysis, the remaining 97.5% of the sample was diluted in a further 3 ml lysis buffer then spun at 14,500 × $g$, 15 min, 4 °C. The supernatants were subjected to end-on rotation at 4 °C with anti-FLAG (Merck, F1804) that was pre-conjugated to Protein A/G beads (Pierce, 88802) for 2 h 4 °C (note: pre-conjugation crosslinked using 20 mM dimethyl pimelimidate (DMP, ThermoFisher) in borate buffer (40 mM boric acid, 40 mM sodium tetraborate decahydrate) 45 min RT). The resulting pull downs were washed four times with lysis buffer before incubation 150 μl 5% SDS, 100 mM TEAB 37 °C, 5 min before the eluate was removed from the beads and processed using S-trap (Protifi) micro protocol according to the manufacturer's instructions. Digested samples were loaded on Evotips and data was acquired on TIMS TOF Pro2 (Bruker) coupled to Evosep One LC system. For the LC separation standard 60SPD 2.3 method was used, separation was performed using EV-1109 column, the column was heated to 40 °C during analyses. TIMS TOF Pro2 was operated in DIA PASEF mode, scan width was set to 100–1700 m/z with the IM 1 K/0 0.6–1.6, ramp and accumulation time was locked at 100 ms. Data acquisition windows are supplied in the Source Data file. Mass spectrometry raw files were analysed using Spectronaut (Biognosis, v18) using the DirectDIA pipeline with standard settings, quantification using at the MS2 level. Human proteome database from Uniprot used as FASTA file (1 protein per gene).

## Plasmid and antibody reagents

Please see Supplementary Table 4 for a full list of plasmids and antibodies used in this study.

## Structural predictions

Predictions were downloaded from AlphaFold2[34,35] and RoseTTAFold (v2)[68] (to predict WT monomeric protein structures according to UniProt canonical isoform assignment, Supplementary Table 4); ColabFold (v1.5.5)[69] (to predict mutant protein structures); and AlphaFold3[70] (to predict dimeric TRIM RING domains with $Zn^{2+}$ co-ordination).

## Statistical analyses

Independent biological repeats of experiments were conducted at least n = 3, as indicated in figure legends. Comparative analyses were performed using two-tailed $t$-tests, with the following asterisk

representations of *p* values: *<0.05, **<0.01, ***<0.001. Standard error of the mean (SEM) is represented as error bars where applicable.

## Data availability

Protein Data Bank accession codes used in this study: 7ZJ3 (TRIM2 RING/UBE2D1-Ub); 5EGG (UBE2D3). The mass spectrometry proteomics data generated in this study have been deposited in the ProteomeXchange Consortium database via the PRIDE partner repository under dataset identifier PXD057364 (interferon-stimulated HEK293T cells ± TRIM6 or TRIM22 overexpression). The processed mass spectrometry proteomics data data are provided in the Source Data file. All the data that support the conclusions in this study, including raw data points and uncropped gels and blots, are available in the Source Data file provided with the article. Source data are provided with this paper.

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

## Acknowledgements

The authors would like to thank Teresa Thurston and Cullum Stones (Imperial College London) for kindly gifting a subset of ptCMV-GFP-TRIM plasmids, Ian Taylor (Francis Crick Institute) for help with SEC-MALLS data acquisition and analysis, and the Cell Services, Proteomics, Structural Biology, and Advanced Light Microscopy Science Technology Platforms at the Francis Crick Institute. This work was supported by the Francis Crick Institute, which receives its core funding from Cancer Research UK (CC2075), the UK Medical Research Council (CC2075), and the Wellcome Trust (CC2075).

## Author contributions

J.D.-F. carried out and analysed experiments; K.A.M. and C.M.-W. carried out experiments; D.E. carried out and analysed NMR experiments; T.A. carried out and analysed mass spectrometry experiments; J.D.-F. and K.R. conceived and designed the study; J.D.-F., D.E., and K.R. wrote the paper.

## Funding

## Competing interests

The authors declare no competing interests.
