## [Transparent Peer Review file · Nature Communications]

Identification of RING E3 pseudoligases in the TRIM protein family

Corresponding Author: Dr Katrin Rittinger

Version 0:

Reviewer comments:

Reviewer #1

(Remarks to the Author)

The study by Dudley-Fraser et al. titled "Identification of RING E3 pseudoligases in the TRIM protein family" sets out an ambitious objective: to perform an unbiased, side-by-side survey of the E3 ligase activity of human TRIM proteins. This is a timely and significant endeavor, as members of this protein family are increasingly recognized for their essential roles in immune signaling and other cellular functions, but mechanistic understanding remains fragmentary. All TRIM proteins share a conserved domain architecture, indicating a common biochemical mechanism underlying their function. Ubiquitin ligase activity is the most likely candidate for this shared functionality, though it has been challenging to establish for some TRIM family members. Identifying outliers that completely lack E3 ubiquitin ligase activity could provide valuable insights and advance the field.

The effort invested in this study is truly remarkable. The sheer number of expression constructs generated is impressive, as is the broad range of experimental techniques employed, spanning cell biology, in-vitro biochemistry, and NMR, complemented by computational structural analysis. However, the study also has significant flaws in experimental design and data interpretation. The main issue is the inherent difficulty of building a critical mass of compelling evidence for negative claims. Experimental failures often occur for trivial reasons, making it challenging to distinguish between genuine evidence of absence and the absence of evidence. In this reviewer's opinion, the lack of E3 ligase activity is not convincingly demonstrated for any of the TRIM proteins (TRIMs 6, 15, 22, and 51) proposed as pseudoligases in this study (see detailed comments below). Moreover, even if the technical issues were resolved, the broader impact of these negative findings on the TRIM field appears limited. Specifically, how does the absence of E3 activity in some TRIM proteins guide future research or enhance understanding of their biochemical roles? These concerns, elaborated below, temper this reviewer's enthusiasm for the manuscript's central claims and overall impact.

1. Inherent limitations in the in-cellulo and in-vitro surveys of E3 activity

- The reliance on ubiquitin staining as evidence for or against E3 activity is problematic. As the authors acknowledge, ubiquitin staining in cells may reflect "TRIM ubiquitination by other E3 ligases present in the cellular milieu". Conversely, "it is largely unknown whether substrate ubiquitination in cells is constant or triggered in response to specific stimuli", so the absence of staining does not confirm the lack of E3 activity, as ubiquitination may be triggered by specific, unknown stimuli.
- The in-vitro experiments with immunoprecipitated TRIM proteins suffer from similar limitations. The lack of activity observed may result from the absence of required cofactors, specialized E2 enzymes, or stimulus-dependent activation. Potential co-immunoprecipitation of other ligases could also confound the interpretation of results.
- The authors suggest that the two surveys may complement each other's weaknesses, but this is not immediately clear. Providing a more explicit explanation of how these approaches complement one another would strengthen the manuscript. Furthermore, the correlation between the two surveys appears modest, raising doubts about whether the datasets effectively identify the best candidates for follow-up.
- The cellular localization data are also limited by potential artifacts from overexpression and GFP fusion. While the authors acknowledge this, the observed discrepancies with previously published results raise questions about the reliability of these findings.

2. Concerns with AlphaFold2 Analysis of RING Structures. While AlphaFold2 has revolutionized structural predictions, its application to evaluating RING domain functionality is problematic for two reasons:

- Limitations in Predicting Small Zinc-Binding Domains: The RING domain is a small zinc-binding motif, making it a borderline case for AlphaFold predictions. Although AlphaFold can reliably predict the core RING structure based on the

characteristic spacing of cysteine and histidine residues, its accuracy diminishes for longer intervening segments, especially the region containing the central alpha helix (helix 2).

- AlphaFold struggles with protein segments that rely on stabilization through oligomerization, which raises concerns about the predicted stability of helices 1 and 3. This limitation affects the interpretation of the dimerization interface and may lead to inaccuracies in modeling these regions. To address this, the authors should consider performing AlphaFold analyses using two separate RING chains to enhance the reliability of predictions for the dimerization interface.

3. Concerns About the Basis for the 'Pseudoligase' Claim. In the second paragraph of the Discussion section, the authors state: "Here we demonstrate that the RING domains of TRIMs 6, 15, 22, and 51 do not catalyse ubiquitination in isolation and, given their structural properties, are unlikely to do so under differing experimental conditions." From this statement, it appears the authors primarily base their 'pseudoligase' claim on experiments with isolated RING domains and associated structural considerations. This approach raises several concerns, which are addressed separately for each TRIM protein below:

- It is unclear why the TRIM15 RING construct was truncated at residue 70. There appear to be 10 additional residues between the RING domain and the B-box, which may contribute to RING dimerization. Assuming the absence of helix 3 solely based on AlphaFold predictions weakens the claim that TRIM15 is a pseudoligase. Furthermore, it is important to recognize that dimerization of isolated RING domains can be weak and often relies on transient proximity mediated by the full-length TRIM protein or its interaction partners. This transient dependency may vary among TRIM RINGs. To address these concerns, the authors should investigate the E3 activity of a tandem TRIM15 RING construct, which could provide a more accurate assessment of its functionality.

- The data for TRIM6 and TRIM22 are the most extensive and compelling in the manuscript, particularly since the authors were able to restore in-cell ubiquitylation and in-vitro activity through specific RING domain mutations. However, the possibility that TRIM6 and TRIM22 require a specialized E2 enzyme not included in the in-vitro experiments remains consistent with the experimental results. Additionally, the authors acknowledge an "apparent inconsistency between the in-vitro data described here and previous cell-based studies." This inconsistency is significant, particularly as prior studies (e.g., PMID: 24882218) demonstrate that TRIM6 ubiquitylation is abolished when a C15A mutation is introduced (Figure 5B). Such evidence strongly supports E3 ligase activity, outweighing the conclusions drawn from the inability to reconstitute activity in vitro.

- The claim that TRIM51 is a pseudoligase is the weakest of the four. Since the authors were unable to generate an isolated RING domain construct for TRIM51, its inclusion in the umbrella statement that "the RING domains of TRIMs 6, 15, 22, and 51 do not catalyse ubiquitination in isolation" is problematic. The remaining data for TRIM51 are insufficient to constitute the critical mass of evidence needed to justify the pseudoligase claim.

4. Concerns About the Use of the Term 'Pseudoligase'. A minor objection to the use of the term 'pseudoligase' arises from the very real possibility, acknowledged by the authors, that heterodimerization among TRIM family members—or specifically between their RING domains—may contribute to their function. Finally, there is a minor objection to the 'pseudoligase' term based on the very real possibility mentioned by the authors that heterodimerization of different TRIM family members or, alternatively, heterodimerization of their RING domains may contribute to their function. In RING heterodimers, the loss of catalytic functionality in one monomer can be compensated by the activity of the other. A well-documented example of this is the BRCA1/BARD1 heterodimer, where functional interplay between the two proteins ensures ubiquitin ligase activity despite one monomer lacking intrinsic catalytic capacity. Based on the criteria put forward in this study, even BRCA1 and BARD1 could potentially be classified as 'pseudoligases,' which highlights the limitations of this designation.

Reviewer #2

(Remarks to the Author)

In this study the authors perform a whole-family analysis of TRIM protein E3 ligase capability, using 293-expressed GFP-tagged TRIM proteins and HA ubiquitin, finding that while many TRIMs are promiscuously active upon overexpression, several are not. Using biochemical and biophysical assays, and structural predictions, the authors map the determinants of 'pseudoligases' and show that these can be fixed by remedying identified absences of known structural and biochemical features of active TRIM RING dimers. The paper demonstrates the utility of using structural predictions to propose novel, testable hypotheses. This is a well-executed study and represents the first whole family analysis of TRIM ligase activity in vitro, nicely complementing the several TRIM family phenotypic screens which have gone before (Uchil 2013, Mandell 2014, Sparrer 2017, Versteeg 2013). The discoveries will be relevant to other protein families characterised by extensive gene duplications. A few minor additions could strengthen the study.

Minor observations:

Fig. 1 is performed rigorously and strengths/limitations of the complementary ubiquitination assays are recognised. The point remains that not knowing which of ~35 E2 enzymes are cofactors for a specific TRIM make screening with a small panel problematic, however the results are clear and select cases of inactivity are further investigated below.

Fig. 1e – should also say 'IP' of GFP TRIM in assay design?

Line 141 – 'it is possible...' is it also possible that lack of activity in vitro is because the E2 cocktail does not contain the necessary E2 for that TRIM.

Sup Fig. 2e would benefit from indicating on the figure itself that samples were taken at either 0 or 120 min, as indicated in figure legend.

Fig. 2b – RHS, TRIM51 should add ' β ' to unfolded 4

Fig. 3 – while the SEC-MALLS and NMR indeed suggest TRIM15 RING is a monomeric species even at high concentration, the lack of activity in vitro (3e) could reflect the limited cohort of E2s tested. Indeed the use of D1 as the E2 in Ub discharge assays (3f) might also reflect incompatibility with D1 rather than inherent lack of ligase activity.

This section could reference the observed activity of TRIM15 in stimulating AP-1 promoters in cells (Uchil and Mothes, JVI,

2013); in this reference, the authors further show that TRIM15 Δ RING is functionally active, supporting the observations made in vitro here.

Fig. 4 – this is a strong dataset, using structural predictions and sequence alignments to 'fix' broken RINGs of TRIM22 and TRIM6; this is particularly interesting given their described roles as E3s in innate immune pathways. As paralogues of TRIM5 α , it serves as a great example of gene duplication preceding evolution of novel function.

In figs 4f and Sup 6c it is unclear why the WT constructs are SUMO tagged while the mutant constructs are not – is there a reason for this? SUMO tagged mutants are also active, or untagged WT remains inactive?

Fig. 5 – could the authors provide a structural model for how TRIM49-51 heterodimers could form via the CC. Do they envisage that the RINGs heterodimerise, as proposed in the discussion. In Sup 7, it looks like IP of FLAG-TRIM49 WT does co-IP GFP-TRIM51 RB, suggesting there is some measurable RING-RING interaction. Is this the case? And in cases where full length heterodimers do form in cells, can the authors speculate on the mechanism of inhibition. Is E2~Ub binding to the active 49 RING impaired by the presence of the inactive 51 RING?

Is there evidence for endogenous TRIM49/51 interaction?

The discussion is very well written. Can the authors suppose that lack of selective pressure results in pseudogenization of duplicated genes, or is there a selective pressure to 'deactivate' prologues in order to generate TRIM49/51-like systems of negative regulation?

The methodology is sound, and sufficient detail is provided to be able to reproduce these experiments.

Reviewer #3

(Remarks to the Author)

In this well-written manuscript, the authors do a systematic analysis of TRIM E3 ligases. They test 68 protein constructs in cells, show their localization. They show that a large number of TRIM proteins do not show autoubiquitylation activity. They identify specific amino acid residues which appear to make the proteins inactive. The authors suggest that these TRIM proteins may be pseudoligases.

This is a very nice paper of an important and timely topic. The authors test the whole family of TRIM proteins and there will be lots of interesting information for the wide ubiquitin community. The shortcomings of their approaches are well discussed. The only criticism would be that the inactive E3-ligases may not do "classical" ubiquitylation, i.e. may use different ubiquitin-like modifiers or perform non-protein ubiquitylation. However, the structural characterization is sound and therefore, the specific amino acids that seem to block E3 ligase activity, appear genuine.

I recommend acceptance of this manuscript after some minor corrections, including a slightly more cautious discussion about the possibilities of different activity as described above.

Version 1:

Reviewer comments:

Reviewer #1

(Remarks to the Author)

The authors have constructively addressed all of my concerns through extensive edits in the revised manuscript and additional data. I now strongly support publication of the study in Nature Communications.

Reviewer #2

(Remarks to the Author)

The authors have taken on board my comments and suggestions and those of the two other reviewers; I am satisfied with their responses. They have performed additional experiments as requested and made necessary amendments to the text and panels. This is a unique and interesting study that adds new understanding to the TRIM family that has gone extensive species-specific gene duplication. The work opens several new avenues for exploration. Not least the question of whether this pseudoligase-ation is a feature of other species, murine trim5 for example.

Minor comment:

Possibly typo in heading of the opening results section (TRIM expression in mammalian cells shows localisation varies within and between classes)

Reviewer #3

(Remarks to the Author)

All good from my side and it can be accepted as is. I want to congratulate the authors to this nice paper.

Response to peer review

We thank the reviewers for taking the time to carefully evaluate our manuscript and for their insightful suggestions regarding experimental approaches and conceptual discussions. We have carried out a number of new experiments to address the concerns raised by the reviewers, which we feel have significantly strengthened the manuscript. Our responses to points raised are described in detail below.

Reviewer #1

The study by Dudley-Fraser et al. titled "Identification of RING E3 pseudoligases in the TRIM protein family" sets out an ambitious objective: to perform an unbiased, side-by-side survey of the E3 ligase activity of human TRIM proteins. This is a timely and significant endeavor, as members of this protein family are increasingly recognized for their essential roles in immune signaling and other cellular functions, but mechanistic understanding remains fragmentary. All TRIM proteins share a conserved domain architecture, indicating a common biochemical mechanism underlying their function. Ubiquitin ligase activity is the most likely candidate for this shared functionality, though it has been challenging to establish for some TRIM family members. Identifying outliers that completely lack E3 ubiquitin ligase activity could provide valuable insights and advance the field.

The effort invested in this study is truly remarkable. The sheer number of expression constructs generated is impressive, as is the broad range of experimental techniques employed, spanning cell biology, in-vitro biochemistry, and NMR, complemented by computational structural analysis. However, the study also has significant flaws in experimental design and data interpretation. The main issue is the inherent difficulty of building a critical mass of compelling evidence for negative claims. Experimental failures often occur for trivial reasons, making it challenging to distinguish between genuine evidence of absence and the absence of evidence. In this reviewer's opinion, the lack of E3 ligase activity is not convincingly demonstrated for any of the TRIM proteins (TRIMs 6, 15, 22, and 51) proposed as pseudoligases in this study (see detailed comments below). Moreover, even if the technical issues were resolved, the broader impact of these negative findings on the TRIM field appears limited. Specifically, how does the absence of E3 activity in some TRIM proteins guide future research or enhance understanding of their biochemical roles? These concerns, elaborated below, temper this reviewer's enthusiasm for the manuscript's central claims and overall impact.

We thank the reviewer for their appreciation of the breadth of techniques employed to answer what is, as they say, a difficult question: how to understand the observations by both ourselves and others in the literature that a subset of TRIM proteins apparently lack ubiquitin E3 ligase activity? This study set out to ascertain the scale of this phenomenon, which we believe has the potential to inform future biochemical and cell-based studies by others into these proteins' functions: whether that be a regulatory role in complexes with ligase-competent TRIMs (e.g. as has been shown from TRIM17/41, PMID 30485814) or in a previously unrecognized ubiquitin-independent function (e.g. Class VI TRIMs: PMID 21531907, 29863470, 31028095). We believe that the biochemical analyses presented here make a valuable contribution to elucidating the functions of these proteins, many of which, as the reviewer mentions, have been shown to have important roles in human disease (e.g. TRIM6 and TRIM22 in infection). Specifically, we hope that our data may prompt researchers to look past previously posited roles as E3 ubiquitin ligases, and encourage discovery of ubiquitin-independent functions.

We have addressed all specific comments raised as detailed below, which we hope alleviates any concerns raised.

1. Inherent limitations in the in-cellulo and in-vitro surveys of E3 activity

- The reliance on ubiquitin staining as evidence for or against E3 activity is problematic. As the authors acknowledge, ubiquitin staining in cells may reflect “TRIM ubiquitination by other E3 ligases present in the cellular milieu”. Conversely, “it is largely unknown whether substrate ubiquitination in cells is constant or triggered in response to specific stimuli”, so the absence of staining does not confirm the lack of E3 activity, as ubiquitination may be triggered by specific, unknown stimuli.

The in-vitro experiments with immunoprecipitated TRIM proteins suffer from similar limitations. The lack of activity observed may result from the absence of required cofactors, specialized E2 enzymes, or stimulus-dependent activation. Potential co-immunoprecipitation of other ligases could also confound the interpretation of results.

We agree with the reviewer on these two points and have attempted to articulate the limitations of these assays in the manuscript. Our ambition was to compare the whole TRIM family in parallel, given that the numerous studies conducted on TRIMs thus far have utilised widely varying reaction conditions and approaches, meaning that they are not directly comparable as a cohort. Therefore, we decided to employ these medium-throughput assays that are widely used throughout the existing body of work on TRIM proteins to provide an initial assessment of ligase activity.

In our follow-up studies on a subset of candidate TRIM proteins, we have attempted to mitigate these limitations using a wider array of conditions including a broader range of E2s and, where possible, recombinant proteins.

We appreciate that we may not have adequately articulated these caveats in the manuscript and have, therefore, expanded our discussion of the limitations of our family-wide assays (page 6, lines 123-133).

- The authors suggest that the two surveys may complement each other's weaknesses, but this is not immediately clear. Providing a more explicit explanation of how these approaches complement one another would strengthen the manuscript. Furthermore, the correlation between the two surveys appears modest, raising doubts about whether the datasets effectively identify the best candidates for follow-up.

We are sorry if we weren't sufficiently explicit on this point. We propose that TRIMs which show activity *in vitro* but not *in cellulo* (n=21: TRIMs 2, 4, 7-10, 17, 18, 34, 38-40, 43, 47, 50, 54, 67-69, 72, and 77) may preferentially ubiquitinate cellular substrates over auto-ubiquitination, an eventuality that is accounted for by complementary *in vitro* assays where substrates are not present. We have now made this explicit in the text (lines 136-137). The candidates we chose to follow up were based on the findings of the *in vitro* ubiquitination assay, given it is a controlled system that is easier to probe experimentally than *in cellulo* assays.

- The cellular localization data are also limited by potential artifacts from overexpression and GFP fusion. While the authors acknowledge this, the observed discrepancies with previously published results raise questions about the reliability of these findings.

We agree that tags on overexpressed proteins are an issue in all cellular studies, but we were encouraged to find that the vast majority of our results are in line with previously published studies, including a recent BioRxiv report which investigated the mesoscale localisation of 72 TRIMs using mEGFP and mCherry-tagged proteins (DOI 10.1101/2025.01.02.630836). Indeed, the localisation 86% of TRIMs examined (58/68) agreed with previous work and the 10 discrepancies observed are potentially attributable to experimental conditions, namely overexpression using different tags (e.g. mCherry, Myc, FLAG, or HA) fused to TRIM proteins (7/10) or staining for endogenous proteins in different cell lines than the one used here (3/10). Therefore, we believe that using a GFP tag in this setting was acceptable, although for completeness we were careful to mention

potential issues in the text (line 101-102). Confirmation of these results by examining the localisation of endogenous proteins would have been desirable, but unfortunately there are no well-validated antibodies available for many TRIM family members.

2. Concerns with AlphaFold2 Analysis of RING Structures. While AlphaFold2 has revolutionized structural predictions, its application to evaluating RING domain functionality is problematic for two reasons:

- Limitations in Predicting Small Zinc-Binding Domains: The RING domain is a small zinc-binding motif, making it a borderline case for AlphaFold predictions. Although AlphaFold can reliably predict the core RING structure based on the characteristic spacing of cysteine and histidine residues, its accuracy diminishes for longer intervening segments, especially the region containing the central alpha helix (helix 2).

We fully agree that predictions generated by AF2 cannot be considered as ‘fact’, instead, we have used them as a guide to suggest mutants to probe potential sequence/structure requirements for ubiquitination activity. We have adjusted our wording in this regard to make it clear that these predictions are guides and we do not consider them as ground-truth (e.g. lines 173, 180, 191).

- AlphaFold struggles with protein segments that rely on stabilization through oligomerization, which raises concerns about the predicted stability of helices 1 and 3. This limitation affects the interpretation of the dimerization interface and may lead to inaccuracies in modeling these regions. To address this, the authors should consider performing AlphaFold analyses using two separate RING chains to enhance the reliability of predictions for the dimerization interface.

This is a good point and, indeed, we have similarly observed an increased confidence for helices 1 and 3 in some TRIM RINGs in dimers as compared to monomers when using AlphaFold. We have now performed RING dimer predictions in addition to modelling Zn²⁺ co-ordination (as flagged by the reviewer, zinc-binding can contribute significantly to proper folding – see also the second bullet point of the reviewer’s comment #3 below), which we include as updated figures throughout the manuscript (Figures 3A, 4B, and 5A, which were predicted as dimers but presented as monomers for ease of viewing).

3. Concerns About the Basis for the ‘Pseudoligase’ Claim. In the second paragraph of the Discussion section, the authors state: “Here we demonstrate that the RING domains of TRIMs 6, 15, 22, and 51 do not catalyse ubiquitination in isolation and, given their structural properties, are unlikely to do so under differing experimental conditions.” From this statement, it appears the authors primarily base their ‘pseudoligase’ claim on experiments with isolated RING domains and associated structural considerations. This approach raises several concerns, which are addressed separately for each TRIM protein below:

- It is unclear why the TRIM15 RING construct was truncated at residue 70. There appear to be 10 additional residues between the RING domain and the B-box, which may contribute to RING dimerization. Assuming the absence of helix 3 solely based on AlphaFold predictions weakens the claim that TRIM15 is a pseudoligase. Furthermore, it is important to recognize that dimerization of isolated RING domains can be weak and often relies on transient proximity mediated by the full-length TRIM protein or its interaction partners. This transient dependency may vary among TRIM RINGs. To address these concerns, the authors should investigate the E3 activity of a tandem TRIM15 RING construct, which could provide a more accurate assessment of its functionality.

We thank the reviewer for their suggestions regarding our TRIM15 data. We apologise for having excluded the additional 10 residues which may have the potential to aid dimerisation, which was an oversight. To assess the contribution of these 10 residues, we have now repeated our experiments using a longer TRIM15 RING construct (residues 2-80). However, these residues did not change the propensity of the RING domain to dimerise or catalyse E3 ubiquitin ligase activity (see updated Fig. 3c-f).

To further substantiate the observed lack of activity, we have additionally produced a tandem RING construct given, as the reviewer states, some TRIM RINGs can show very weak but functionally significant homo-dimerisation. For example, although TRIM25 RING domain has barely detectable self-association and is largely monomeric in solution when analysed by SEC-MALLS, it exhibits E3 ubiquitin ligase activity in *in vitro* assays that is significantly enhanced in a tandem RING construct, (PMID: 27154206). To analyse this eventuality for TRIM15, we performed both auto-ubiquitination and ubiquitin discharge assays using a tandem fusion construct of TRIM15 (2 x domains of residues 2-80 connected by a Ser residue, as previously described for TRIM25 and TRIM3 (PMID: 36481767). These experiments showed that a forced 'dimer'-like protein still does not exhibit any ubiquitin ligase activity with UBE2D1 or UBE2N/V2 (see new Supplementary Fig.s 4c-e, and lines 208-215). Supplementary Fig. 4f shows the AlphaFold3 modelling of a Zn²⁺-co-ordinated TRIM15 tandem construct, which predicts five entirely distinct conformers, suggesting that this construct likely does not adopt a stable conformation that is conducive to ligase activity. Together, these data suggest that TRIM15 has no tendency to dimerise and formation of an active dimer cannot be induced in a tandem RING construct.

- The data for TRIM6 and TRIM22 are the most extensive and compelling in the manuscript, particularly since the authors were able to restore in-cell ubiquitylation and in-vitro activity through specific RING domain mutations. However, the possibility that TRIM6 and TRIM22 require a specialized E2 enzyme not included in the in-vitro experiments remains consistent with the experimental results. Additionally, the authors acknowledge an "apparent inconsistency between the in-vitro data described here and previous cell-based studies." This inconsistency is significant, particularly as prior studies (e.g., PMID: 24882218) demonstrate that TRIM6 ubiquitylation is abolished when a C15A mutation is introduced (Figure 5B). Such evidence strongly supports E3 ligase activity, outweighing the conclusions drawn from the inability to reconstitute activity in vitro.

We agree that it is difficult to reconcile the data presented here with some of those in the literature. However, as the reviewer mentions in their point #1, studies such as the one cited (PMID 24882218) employ *in cellulo* ubiquitination assays and/or protein immunoprecipitated from cells that must be considered in light of their limitations, such as the presence of confounding ligases, as discussed above. Moreover, a C15A mutation would abolish not only ubiquitin ligase activity, but also possibly the proper folding of the domain, given this residue is required to co-ordinate zinc. Therefore this mutation offers the potential to disrupt domain folding and hence any potential domain function, whether ubiquitin-related or not.

Furthermore, TRIMs 6 and 22 have been primarily studied in the context of infection, where it is possible that, because of the experimental conditions, TRIM21 expression is induced. As we have shown in Supplementary Figure 5, TRIM21 expression induced by inflammatory signalling is a significant concern when carrying out ubiquitination analyses following immunoprecipitation as TRIM21 binds directly to antibodies (as well-described in the work of Leo James and colleagues – e.g. PMIDs: 23455675, 21045130, 18420815), and may hence contribute to any observed E3 ligase activity.

Importantly however, we believe that the ability to use mutagenesis to 'fix' broken RINGs

of TRIM22 and TRIM6, based on structural predictions and sequence alignments further supports our conclusion, as also highlighted by reviewer 2. Nevertheless, we appreciate that we might be missing a signal or co-factor and have therefore attempted to align our analyses of TRIMs 6 and 22 *in vitro* with existing data regarding their function in cells by adding text to clarify that there is a possibility that these proteins could catalyse ubiquitination under hitherto unknown, specific conditions (see lines 326-327).

- The claim that TRIM51 is a pseudoligase is the weakest of the four. Since the authors were unable to generate an isolated RING domain construct for TRIM51, its inclusion in the umbrella statement that “the RING domains of TRIMs 6, 15, 22, and 51 do not catalyse ubiquitination in isolation” is problematic. The remaining data for TRIM51 are insufficient to constitute the critical mass of evidence needed to justify the pseudoligase claim.

We agree with the reviewer that it is unfortunate that we cannot develop our hypothesis regarding TRIM51 by testing it in isolation *in vitro* due to technical difficulties with recombinant protein purification (TRIM49 and TRIM51 RING, RING-B box, and RING-B box-coiled coil constructs all proved insoluble, despite our best efforts). However, given that we can swap the β 3- β 4 region from TRIM51 into TRIM49 and thereby render active TRIM49 inactive, while the reverse swap restored some activity in TRIM51, we feel that this provides strong support for our suggestion that TRIM51 is likely unable to interact with an E2~Ub conjugate. We have modified our wording to soften our claim in this regard (see lines 375-376).

4. Concerns About the Use of the Term ‘Pseudoligase’. A minor objection to the use of the term ‘pseudoligase’ arises from the very real possibility, acknowledged by the authors, that heterodimerization among TRIM family members—or specifically between their RING domains—may contribute to their function. Finally, there is a minor objection to the ‘pseudoligase’ term based on the very real possibility mentioned by the authors that heterodimerization of different TRIM family members or, alternatively, heterodimerization of their RING domains may contribute to their function. In RING heterodimers, the loss of catalytic functionality in one monomer can be compensated by the activity of the other. A well-documented example of this is the BRCA1/BARD1 heterodimer, where functional interplay between the two proteins ensures ubiquitin ligase activity despite one monomer lacking intrinsic catalytic capacity. Based on the criteria put forward in this study, even BRCA1 and BARD1 could potentially be classified as ‘pseudoligases,’ which highlights the limitations of this designation.

We based our use of the ‘pseudoenzyme’ terminology on the literature-reported use of this term for other enzymes, particularly kinases (e.g. Sheetz & Lemmon, PMID: 35585008). This definition would include proteins that are catalytically dead themselves but form a functional regulatory (activating or repressive) complex with their active counterparts.

In their 2017 publication, Murphy *et al* (PMID: 28787627) state: ‘A conceptually simple mechanism that helps explain the prevalence of pseudoenzymes in biology is the finding that, upon binding, a pseudoenzyme can often impact upon the catalytic activity of a conventional, often related, enzyme (or non-enzyme) protein. The best-characterised examples of such allostery involve pseudoenzymes regulating a structurally related enzyme counterpart’. They cite allosteric activatory mechanisms including: a) allosteric stabilisation of the active enzyme’s active site, b) stabilisation of an additional interaction partner required for catalytic activity of the active kinase, and c) promotion of active enzyme multimerisation. Sheetz & Lemmon state this in terms of kinases, for example, where ErbB3 is an inactive pseudokinase in isolation, but allosterically activates EGFR kinase activity. Indeed, others have previously described BARD1, MDMX1, and PCGF-1 as ‘pseudo-E3 Ub ligases’ (PMID: 28408493).

Therefore, we consider that the data presented here support the definition of these TRIM proteins as pseudoenzymes, i.e. pseudoligases, irrespective the potential for heterodimerisation. We have clarified this in our discussion of the pseudoligase concept (line 374-375).

Reviewer #2

In this study the authors perform a whole-family analysis of TRIM protein E3 ligase capability, using 293-expressed GFP-tagged TRIM proteins and HA ubiquitin, finding that while many TRIMs are promiscuously active upon overexpression, several are not. Using biochemical and biophysical assays, and structural predictions, the authors map the determinants of 'pseudoligases' and show that these can be fixed by remedying identified absences of known structural and biochemical features of active TRIM RING dimers. The paper demonstrates the utility of using structural predictions to propose novel, testable hypotheses. This is a well-executed study and represents the first whole family analysis of TRIM ligase activity *in vitro*, nicely complementing the several TRIM family phenotypic screens which have gone before (Uchil 2013, Mandell 2014, Sparrer 2017, Versteeg 2013). The discoveries will be relevant to other protein families characterised by extensive gene duplications. A few minor additions could strengthen the study.

We would like to thank this reviewer for their supportive comments on our study design and findings, as well as for their suggestions, which we believe have strengthened the manuscript.

Minor observations:

Fig. 1 is performed rigorously and strengths/limitations of the complementary ubiquitination assays are recognised. The point remains that not knowing which of ~35 E2 enzymes are cofactors for a specific TRIM make screening with a small panel problematic, however the results are clear and select cases of inactivity are further investigated below.

We thank the reviewer for their consideration of the assay design and discussion. We agree that it would be ideal to include all E2 enzymes in our screen, which unfortunately was experimentally not feasible. For this reason we have compared *in cellulo* assays which should contain all E2s expressed in a given cell type, with *in vitro* assays to allow for more controlled assay conditions. In addition, we attempted to mitigate for this in follow-up studies with additional E2 enzymes (e.g. a further four E2s were trialled with TRIM15) and have discussed this limitation where appropriate (e.g. see lines 130, 145, 326).

Fig. 1e – should also say 'IP' of GFP TRIM in assay design?

We thank the reviewer for spotting this oversight, which we have corrected.

Line 141 – 'it is possible...' is it also possible that lack of activity *in vitro* is because the E2 cocktail does not contain the necessary E2 for that TRIM.

We agree that this is possible. Therefore, in our follow-up studies, we used all of the E2s that had been reported in previous studies to function with the respective TRIMs: TRIM15 with UBE2N/V2 (PMID: 34497368); TRIM22 with UBE2D family members (PMID: 18656448); TRIM6 with UBE2K (PMID: 24882218); and TRIMs 6 and 22 both with UBE2N/V2, based on the mechanisms of closely related class IV family members TRIM5 and TRIM21 (e.g. PMID: 31582740). To highlight that alternative E2s may be used by these TRIMs we have altered the text accordingly (now line 145). Please also see our reply regarding E2s used in the previous comment.

Sup Fig. 2e would benefit from indicating on the figure itself that samples were taken at either 0 or 120 min, as indicated in figure legend.

We thank the reviewer for identifying this missing labelling, which we have now added.

Fig. 2b – RHS, TRIM51 should add 'β' to unfolded 4

We apologise for this typo, which we have corrected.

Fig. 3 – while the SEC-MALLS and NMR indeed suggest TRIM15 RING is a monomeric species even at high concentration, the lack of activity in vitro (3e) could reflect the limited cohort of E2s tested. Indeed the use of D1 as the E2 in Ub discharge assays (3f) might also reflect incompatibility with D1 rather than inherent lack of ligase activity. This section could reference the observed activity of TRIM15 in stimulating AP-1 promoters in cells (Uchil and Mothes, JVI, 2013); in this reference, the authors further show that TRIM15 Δ RING is functionally active, supporting the observations made in vitro here.

We thank the reviewer for their consideration of our TRIM15 data. We have now included the reference for Uchil & Mothes, which we agree offers interesting supporting data to the hypothesis of TRIM15 as a 'pseudoligase' with alternative functions that we propose here. We have included this point in the discussion of our results (lines 215-222).

We now also explicitly say that ubiquitin assays were performed with a limited range of E2s (line 145), albeit including the E2 that has previously been described to function with TRIM15 (UBE2N/V2, PMID: 34497368), which we have added to this section of the results (line 202).

Fig. 4 – this is a strong dataset, using structural predictions and sequence alignments to 'fix' broken RINGs of TRIM22 and TRIM6; this is particularly interesting given their described roles as E3s in innate immune pathways. As paralogues of TRIM5alpha, it serves as a great example of gene duplication preceding evolution of novel function.

We are delighted that the reviewer has noted this idea, which we believe is an exciting avenue for future research.

In figs 4f and Sup 6c it is unclear why the WT constructs are SUMO tagged while the mutant constructs are not – is there a reason for this? SUMO tagged mutants are also active, or untagged WT remains inactive?

There was no scientific reason why these constructs were SUMO tagged, rather simply that these were the constructs available at the time of the experiment. We apologise for any confusion this may have caused. These assays have now been repeated with untagged TRIM6 and TRIM22 and are shown in Fig 4f and Supplementary Fig 5c. This has not changed the outcome. We note that the mutant proteins run at a slightly lower molecular weight than their wild-type counterparts.

Fig. 5 – could the authors provide a structural model for how TRIM49-51 heterodimers could form via the CC. Do they envisage that the RINGs heterodimerise, as proposed in the discussion.

In Sup 7, it looks like IP of FLAG-TRIM49 WT does co-IP GFP-TRIM51 RB, suggesting there is some measurable RING-RING interaction. Is this the case? And in cases where full length heterodimers do form in cells, can the authors speculate on the mechanism of inhibition. Is E2~Ub binding to the active 49 RING impaired by the presence of the inactive 51 RING? Is there evidence for endogenous TRIM49/51 interaction?

We thank the reviewer for their comments on Figure 5 and Supplementary Figure 7.

We believe that the band just below the 55 kDa marker in lane 9 of the right-hand blot against GFP (i.e. IP of FLAG-TRIM49 WT, co-expression of GFP-TRIM51 RB domains) in Supplementary Figure 7 of the original submission is a non-specific band of a very similar, although not identical, molecular weight in the GFP IP. This band can be seen to a lesser extent in the lanes 10-13, but particularly also in lane 14 (IP of FLAG-TRIM49 WT, co-expression of GFP-TRIM51 SPRY domain – see blots for ‘n=2’ in Revision Figure 1, below).

We also do not observe any co-IP between GFP-TRIM51 WT and FLAG-TRIM49 RB (i.e. lane 2 of the right-hand blot against FLAG), supporting the idea that RING-RING heterodimerisation does not primarily drive this interaction. We have now reduced the amount of antibody used in the IP and a new blot is included to illustrate this point (Revision Figure 1 ‘n=1’ and Supplementary Fig. 7). Therefore, we do not currently have any evidence that the RING domains of TRIM49 and TRIM51 interact in the absence of the remainder of the proteins.

Revision Figure 1: Comparison of TRIM49/TRIM51 co-IP Repeats

n=3 repeats of the experiment shown in Supplementary Figure 7 to identify the domains of TRIM49 and TRIM51 that interact, showing anti-GFP blots of whole cell lysates (left) vs IP samples (right). Pink dashed line represents the comparison of the RB construct band between the two blots at the equivalent molecular weight. Please note, ‘n=3’ samples are run in a modified order (‘CC’ and ‘CC-SPRY’ are swapped, relative to ‘n=1’ and ‘n=2’).

Unfortunately, despite our best efforts we have been unable to produce recombinant TRIM49 or TRIM51 RING domains. This means that we are unable to assess whether TRIM51 disrupts TRIM49 interaction with the E2~Ub conjugate. We do hope that future studies may be able to better understand this interplay.

We were also keen to identify cell lines that express TRIM49 and TRIM51 and subsequently explore their potential endogenous interaction. However, after an exhaustive search of manufacturers, we were unable to identify an antibody with an epitope that could distinguish TRIM49 from TRIM51, given their high degree of homology. Therefore, at this point we are unable to further explore if TRIM49 and TRIM51 interact endogenously.

We have now performed AlphaFold3 predictions of TRIM49-51 heterotetramers, to model how the coiled-coil and potentially RING domains might interact. Predictions were set up either as two copies of each TRIM sequence or as 4 individual sequences. This generated very different models: either heterodimers which associated into a heterotetramer (Revision Figure 2a) or a heterotetramer where each coiled coil region folds back on itself and the RINGs mediate assembly through heterodimerisation. Both models have very high Predicted Alignment Errors (PAE) indicating low reliability of the predictions.

Importantly, none of these predictions make biological sense. All TRIM proteins that have been structurally characterised thus far show a conserved mode of homodimerisation via their coiled-coil regions an interaction that buries an extensive interface, and is unlikely to be disrupted once formed. Given this, as well as a lack of convergence on a single conformer, we did not consider it appropriate to include the predictions in the manuscript or to speculate on potential structural models without further experimental data.

Revision Figure 2: TRIM49-TRIM51 Interaction Structural Models by AlphaFold3

The discussion is very well written. Can the authors suppose that lack of selective pressure results in pseudogenization of duplicated genes, or is there a selective pressure to 'deactivate' prologues in order to generate TRIM49/51-like systems of negative regulation? The methodology is sound, and sufficient detail is provided to be able to reproduce these experiments.

We thank the reviewer for their thoughts. We are similarly intrigued by the idea of evolution and selective pressure. Indeed, class IV TRIMs in particular, which are located in the MHC locus (PMID: 18673550), have a particular susceptibility to duplication and we see TRIMs of this class enriched as 'pseudoligase' TRIMs. We are lacking the genetic data to support this notion, but it is tempting to suggest that gene duplication resulting in an active ligase mutation to become inactive may offer an organism a novel selective advantage to its environment (through mechanisms unknown). Therefore, we speculate that the potential selective pressure would be 'positive' rather than 'negative' – to offer an advantage despite a lack of selective pressure rather than survival under stressors (i.e. the first possibility proposed by the reviewer). We would be keen to see this as the subject of future work by relevant specialists.

Reviewer #3

In this well-written manuscript, the authors do a systematic analysis of TRIM E3 ligases. They test 68 protein constructs in cells, show their localization. They show that a large number of TRIM proteins do not show autoubiquitylation activity. They identify specific amino acid residues which appear to make the proteins inactive. The authors suggest that these TRIM proteins may be pseudoligases. This is a very nice paper of an important and timely topic. The authors test the whole family of TRIM proteins and there will be lots of interesting information for the wide ubiquitin community. The shortcomings of their approaches are well discussed.

We would like to thank the reviewer for their supportive assessment of our manuscript and hope that it may be of use to the ubiquitin community, as they suggest.

The only criticism would be that the inactive E3-ligases may not do "classical" ubiquitylation, i.e. may use different ubiquitin-like modifiers or perform non-protein ubiquitylation. However, the structural characterization is sound and therefore, the specific amino acids that seem to block E3 ligase activity, appear genuine. I recommend acceptance of this manuscript after some minor corrections, including a slightly more cautious discussion about the possibilities of different activity as described above.

This is a very good point, given the expanding repertoire of hitherto unappreciated ubiquitination modes, which we thank the reviewer for raising. We have expanded on the possibility of non-canonical ubiquitination and ubiquitin-like modifiers in our discussion (see lines 78, 376, and 413-415).